# Causal Effect Inference for Structured Treatments

**Jean Kaddour**[*]
Centre for Artificial Intelligence
University College London

**Yuchen Zhu**
Centre for Artificial Intelligence
University College London

**Qi Liu**
Department of Computer Science
University of Oxford

**Matt J. Kusner**
Centre for Artificial Intelligence
University College London

**Ricardo Silva**
Department of Statistical Science
University College London

## Abstract

We address the estimation of conditional average treatment effects (CATEs) for structured treatments (e.g., graphs, images, texts). Given a weak condition on the effect, we propose the *generalized Robinson decomposition*, which (i) isolates the causal estimand (reducing regularization bias), (ii) allows one to plug in arbitrary models for learning, and (iii) possesses a quasi-oracle convergence guarantee under mild assumptions. In experiments with small-world and molecular graphs we demonstrate that our approach outperforms prior work in CATE estimation.

## 1 Introduction

Estimating feature-level causal effects, so-called *conditional average treatment effects* (CATEs), from observational data is a fundamental problem across many domains. Examples include understanding the effects of non-pharmaceutical interventions on the transmission of COVID-19 in a specific region [12], how school meal programs impact child health [13], and the effects of chemotherapy drugs on cancer patients [52]. Supervised learning methods face two challenges in such settings: (i) *missing interventions*, the fact that we only observe one treatment for each individual means models must extrapolate to new treatments without access to ground truth, and (ii) *confounding factors* that affect both treatment assignment and the outcome means that extrapolation from observation to intervention requires assumptions. Many approaches have been proposed to overcome these issues [1, 2, 3, 4, 5, 6, 7, 9, 10, 15, 18, 19, 21, 22, 23, 25, 27, 29, 33, 39, 41, 42, 45, 52, 56, 57, 60, 64, 67].

In many cases, treatments are naturally *structured*. For instance, a drug is commonly represented by its molecular structure (graph), the nutritional content of a meal as a food label (text), and geographic regions affected by a new policy as a map (image). Taking this structure into account can provide several advantages: (i) higher data-efficiency, (ii) capability to work with many treatments, and (iii) generalizing to unseen treatments during test time. However, the vast majority of prior work operates on either binary or continuous scalar treatments (structured treatments are rarely considered, a notable exception to this trend is Harada & Kashima [16] which we describe in Section 2).

To estimate CATEs with structured interventions, our contributions include:

- **Generalized Robinson decomposition (GRD):** A generalization of the Robinson decomposition [47] to treatments that can be vectorized as a continuous embedding. This GRD reveals a learnable

---

[*]Correspondence to jean.kaddour.20@ucl.ac.uk

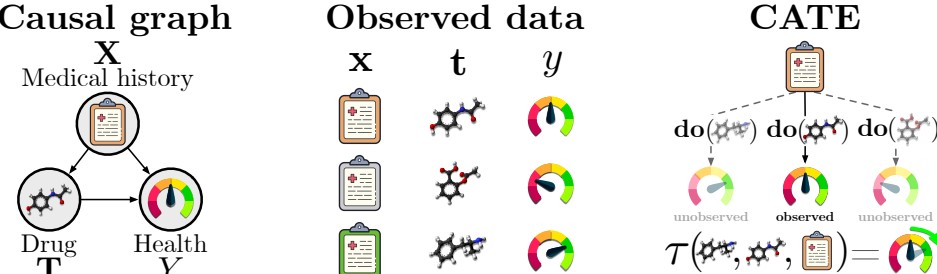

Figure 1: **Illustration of CATE estimation with structured treatments (e.g., molecular graphs).** *Left*: Problem setup with features $\mathbf{X}$, treatment $\mathbf{T}$, and outcome $Y$. *Center*: Observations the estimator has access to, typically containing only one outcome per individual. *Right*: The CATE is the difference between the expected outcomes given a fixed individual and a pair of treatments.

pseudo-outcome target that isolates the causal component of the observed signal by partialling out confounding associations. Further, it allows one to learn the nuisance and target functions using any supervised learning method, thus extending recent work on *plug-in estimators* [42, 29].

- **Quasi-oracle convergence guarantee:** A result that shows that given access to estimators of certain nuisance functions, as long as the estimates converge at an $O(n^{-1/4})$ rate, the target estimator for the CATE achieves the same error bounds as an oracle who has ground-truth knowledge of both nuisance components, the propensity features, and conditional mean outcome.

- **Structured Intervention Networks (SIN)**: A practical algorithm using GRD, representation learning, and alternating gradient descent. Our PyTorch [43] implementation is online.[2]

- **Evaluation metrics** designed for structured treatments. Since previous evaluation protocols of CATE estimators have mostly focused on binary or scalar-continuous treatment settings, we believe that our proposed evaluation metrics can be useful for comparing future work.

- **Experimental results** with graph treatments in which SIN outperforms previous approaches.

## 2 Related Work

Closest to our work is GraphITE [16], a method that learns representations of graph interventions for CATE estimation. They propose to minimize prediction loss plus a regularization term that aims to control for confounding based on the Hilbert-Schmidt Independence Criterion (HSIC) [14]. This technique suffers from two drawbacks: (i) the HSIC requires multiplication of kernel matrices and scales quadratically in the batch size; (ii) selecting the HSIC kernel hyper-parameter is not straightforward, as ground-truth CATEs are never observed, and empirical loss does not bound CATE estimation error [1]. We discuss other related work not on structured treatments in Appendix A.

## 3 Preliminaries

### 3.1 Conditional Average Treatment Effects (CATEs)

Imagine a dataset where each example $(\mathbf{x}_i, \mathbf{t}_i, y_i) \in \mathcal{D}$ represents a hospital patient's medical history record $\mathbf{x}_i$, prescribed drug treatment $\mathbf{t}_i$, and health outcome $y_i$, as illustrated in Figure 1 (*Center*). Further, we wish to understand how changing the treatment changes a patient's health outcome. The CATE, $\tau\left(\mathbf{t}', \mathbf{t}_i, \mathbf{x}_i\right)$, describes the expected change in outcome for individuals with history $\mathbf{x}_i$, when treatment $\mathbf{t}_i$ is replaced by $\mathbf{t}'$, depicted in Figure 1 (*Right*). In real-world scenarios, we only observe one outcome for each patient at one treatment level. Further, the patient's pre-treatment health conditions $\mathbf{x}_i$ influence both the doctor's treatment prescription and outcome, thereby *confounding* the effect of the treatment on the outcome.

Formally, we have the dataset $\mathcal{D} = \left\{(\mathbf{x}_i, \mathbf{t}_i, y_i)\right\}_{i=1}^{n}$ sampled from a joint distribution $p\left(\mathbf{X}, \mathbf{T}, Y\right)$, where $Y = f\left(\mathbf{X}, \mathbf{T}\right) + \varepsilon$, as depicted in Figure 1 (*Left*). We define the causal effect of fixing

---

[2]https://github.com/JeanKaddour/SIN

*treatment* variable $\mathbf{T} \in \mathcal{T}$ to a value $\mathbf{t}$ on *outcome* variable $Y \in \mathbb{R}$ using the do-operator [44] as $\mathbb{E}\left[Y \mid \mathrm{do}(\mathbf{T} = \mathbf{t})\right]$. Crucially, this estimate differs from the conditional expectation $\mathbb{E}\left[Y \mid \mathbf{T} = \mathbf{t}\right]$ in that it describes the effect of an external entity *intervening* on $\mathbf{T}$ by fixing it to a value $\mathbf{t}$ (removing the edge $\mathbf{X} \to \mathbf{T}$). We further condition on pre-treatment *covariates* $\mathbf{X}$ to define the conditional causal estimand $\mathbb{E}\left[Y \mid \mathbf{X} = \mathbf{x}, \mathrm{do}(\mathbf{T} = \mathbf{t})\right]$. The *conditional average treatment effect* (CATE) is the difference between expected outcomes at different treatment values $\mathbf{t}, \mathbf{t}'$ for given covariates $\mathbf{x}$,

$$\tau(\mathbf{t}', \mathbf{t}, \mathbf{x}) \triangleq \underbrace{\mathbb{E}\left[Y \mid \mathbf{X} = \mathbf{x}, \mathrm{do}(\mathbf{T} = \mathbf{t}')\right]}_{=:\mu_{\mathbf{t}'}(\mathbf{x})} - \underbrace{\mathbb{E}\left[Y \mid \mathbf{X} = \mathbf{x}, \mathrm{do}(\mathbf{T} = \mathbf{t})\right]}_{=:\mu_{\mathbf{t}}(\mathbf{x})}, \tag{1}$$

where $\mu_{\mathbf{t}}(\mathbf{x})$ is defined as the *expected outcome* for a covariate vector $\mathbf{x}$ under treatment $\mathbf{t}$.

Because we do not observe both treatments $\mathbf{t}, \mathbf{t}'$ for a single covariate $\mathbf{x}$, we need to make assumptions that allow us to identify the CATE from observational data.

**Assumption 1.** *(Unconfoundedness) There are no confounders of the effect between $\mathbf{T}$ and $Y$ beyond $\mathbf{X}$. Therefore, $Pr\left(Y \le y \mid \mathbf{x}, do(\mathbf{t})\right) = Pr\left(Y \le y \mid \mathbf{x}, \mathbf{t}\right)$, for all $(\mathbf{x}, \mathbf{t}, y)$.*

**Assumption 2.** *(Overlap) It holds that $0 < p\left(\mathbf{t} \mid \mathbf{x}\right) < 1$, for all $(\mathbf{x}, \mathbf{t})$.*

Assumption 2 means that all sub-populations have some probability of receiving any value of treatment (otherwise, some $\tau(\mathbf{t}', \mathbf{t}, \mathbf{x})$ may be undefined or impossible to estimate.) These assumptions allow us to estimate the causal quantity $\tau(\mathbf{t}', \mathbf{t}, \mathbf{x})$ through statistical estimands:

$$\tau\left(\mathbf{t}', \mathbf{t}, \mathbf{x}\right) = \mu_{\mathbf{t}'}\left(\mathbf{x}\right) - \mu_{\mathbf{t}}\left(\mathbf{x}\right) = \mathbb{E}\left[Y \mid \mathbf{X} = \mathbf{x}, \mathbf{T} = \mathbf{t}'\right] - \mathbb{E}\left[Y \mid \mathbf{X} = \mathbf{x}, \mathbf{T} = \mathbf{t}\right]. \tag{2}$$

While one can model $\mu_t(\mathbf{x})$ with regression models, such approaches suffer from bias [9, 26, 29] due to two factors: (i) associations between $\mathbf{X}$ and $\mathbf{T}$, due to confounding, makes it hard to identify the distinct contributions of $\mathbf{X}$ and $\mathbf{T}$ on $Y$, and (ii) regularization for predictive performance can harm effect estimation. Mitigating these biases relies on exposing and removing *nuisance components*. This transforms the optimization into a (regularized) regression problem that isolates the causal effect.

## 3.2 Robinson Decomposition

One way to formulate such nuisance components is via the *Robinson decomposition* [47]. Originally a reformulation of the CATE for binary treatments, it was used by the *R-learner* [42] to construct a plug-in estimator. The R-learner exploits the decomposition by partialling out the confounding of $\mathbf{X}$ on $\mathbf{T}$ and $Y$. It also isolates the CATE, thereby removing regularization bias.

Let the treatment variable be $T \in \{0, 1\}$ and the outcome model $p\left(y \mid \mathbf{x}, \mathbf{t}\right)$ parameterized as

$$Y = f(\mathbf{X}, T) + \varepsilon \equiv \mu_0(\mathbf{X}) + T \times \tau_b(\mathbf{X}) + \varepsilon, \tag{3}$$

where we define error term $\varepsilon$ such that $\mathbb{E}\left[\varepsilon \mid \mathbf{x}, \mathbf{t}\right] = \mathbb{E}\left[\varepsilon \mid \mathbf{x}\right] = 0$, and $\tau_b\left(\mathbf{x}\right) \triangleq \tau\left(1, 0, \mathbf{x}\right)$.

Define the *propensity score* [48] $e\left(\mathbf{x}\right) \triangleq p\left(T = 1 \mid \mathbf{x}\right)$ and the *conditional mean outcome* as

$$m\left(\mathbf{x}\right) \triangleq \mathbb{E}\left[Y \mid \mathbf{x}\right] = \mu_0\left(\mathbf{x}\right) + e\left(\mathbf{x}\right)\tau_b\left(\mathbf{x}\right). \tag{4}$$

From model (3) and the previous definitions, it follows that

$$Y - m\left(\mathbf{X}\right) = \left(T - e\left(\mathbf{X}\right)\right)\tau_b\left(\mathbf{X}\right) + \varepsilon, \tag{5}$$

allowing us to define the estimator

$$\widehat{\tau}_b\left(\cdot\right) = \underset{\tau_b}{\arg\min}\left\{\frac{1}{n}\sum_{i=1}^{n}\left(\tilde{y}_i - \tilde{t}_i \times \tau_b\left(\mathbf{x}_i\right)\right)^2 + \Lambda\left(\tau_b\left(\cdot\right)\right)\right\}, \tag{6}$$

where $\tilde{y}_i \triangleq y_i - \widehat{m}\left(\mathbf{x}_i\right)$ and $\tilde{t}_i \triangleq t_i - \widehat{e}\left(\mathbf{x}_i\right)$ are pseudo-data points defined through estimated nuisance functions $\widehat{m}(\cdot), \widehat{e}(\cdot)$, which can be learned separately with any supervised learning algorithm.

# 4 The Generalized Robinson Decomposition

Our goal is to estimate the CATE $\tau(\mathbf{t}', \mathbf{t}, \mathbf{x})$ for structured interventions $\mathbf{t}', \mathbf{t}$ (e.g., graphs, images, text) while accounting for the confounding of $\mathbf{X}$ on $\mathbf{T}$ and $Y$. Inspired by the Robinson decomposition, which has enabled flexible CATE estimation for binary treatments [6, 9, 33, 42], we propose the *Generalized Robinson Decomposition* from which we extract a pseudo-outcome that targets the causal effect. We demonstrate the usefulness of this decomposition from both a theoretical view (quasi-oracle convergence rate in Section 4.2) and practical view (*Structured Intervention Networks* in Section 5). For details on its motivation and derivation, we refer the reader to Appendix B.

## 4.1 Generalizing the Robinson Decomposition

To generalize the Robinson decomposition to structured treatments, we introduce two concepts: (a) we assume that the causal effect is a *product effect*: the outcome function $f^*(\mathbf{X}, \mathbf{T})$ can be written as an inner product of two separate functionals, one over the covariates and one over the treatment, and (b) *propensity features*, which partial out the effects from the covariates on the treatment features. Similar techniques have been previously shown to add to the robustness of estimation [9, 42].

**Assumption 3.** *(Product effect) We consider the following partial parameterization of $p(y \mid \mathbf{x}, \mathbf{t})$,*

$$Y = g\left(\mathbf{X}\right)^\top h\left(\mathbf{T}\right) + \varepsilon, \tag{7}$$

*where $g : \mathcal{X} \to \mathbb{R}^d, h : \mathcal{T} \to \mathbb{R}^d$ and $\mathbb{E}[\varepsilon \mid \mathbf{x}, \mathbf{t}] = \mathbb{E}\left[\varepsilon \mid \mathbf{x}\right] = 0,$ for all $(\mathbf{x}, \mathbf{t}) \in \mathcal{X} \times \mathcal{T}.$*

This assumption is mild, as we can formally justify its universality. The following asserts that provided we allow the dimensionality of $g$ and $h$ to grow, we may approximate any arbitrary bounded continuous functions in $\mathcal{C}\left(\mathcal{X} \times \mathcal{T}\right)$ where $\mathcal{X} \times \mathcal{T}$ is compact.

**Proposition 1.** *(Universality of product effect) Let $\mathcal{H}_{\mathcal{X} \times \mathcal{T}}$ be a Reproducing Kernel Hilbert Space (RKHS) on the set $\mathcal{X} \times \mathcal{T}$ with universal kernel $k$. For any $\delta > 0$, and any $f \in \mathcal{H}_{\mathcal{X} \times \mathcal{T}}$, there is a $d \in \mathbb{N}$ such that there exist two $d$-dimensional vector fields $g : \mathcal{X} \to \mathbb{R}^d$ and $h : \mathcal{T} \to \mathbb{R}^d$, where $\|f - g^\top h\|_{L_2(P_{\mathcal{X} \times \mathcal{T}})} \leq \delta$. (Proof in Appendix C)*

This assumption allows us to simplify the expression of the CATE for treatments $\mathbf{t}', \mathbf{t}$, given $\mathbf{x}$,

$$\tau\left(\mathbf{t}', \mathbf{t}, \mathbf{x}\right) = g\left(\mathbf{x}\right)^\top \left(h\left(\mathbf{t}'\right) - h\left(\mathbf{t}\right)\right). \tag{8}$$

Define *propensity features* $e^h\left(\mathbf{x}\right) \triangleq \mathbb{E}\left[h\left(\mathbf{T}\right) \mid \mathbf{x}\right]$ and $m\left(\mathbf{x}\right) \triangleq \mathbb{E}\left[Y \mid \mathbf{x}\right] = g\left(\mathbf{x}\right)^\top e^h\left(\mathbf{x}\right).$

Following the same steps as in Section 3.2, the Generalized Robinson Decomposition for eq. (7) is

$$\boxed{Y - m\left(\mathbf{X}\right) = g\left(\mathbf{X}\right)^\top \left(h\left(\mathbf{T}\right) - e^h\left(\mathbf{X}\right)\right) + \varepsilon.} \tag{9}$$

Given nuisance estimates $\widehat{m}(\cdot), \widehat{e}^h(\cdot)$, we can use this decomposition to derive an optimization problem for $h(\cdot), g(\cdot)$ (note $\widehat{e}^h(\cdot)$ implicitly depends on $h(\cdot)$, we address this dependence in Section 5).

$$\widehat{g}\left(\cdot\right), \widehat{h}\left(\cdot\right) \triangleq \underset{g,h}{\arg\min} \left\{ \frac{1}{n} \sum_{i=1}^n \left(Y_i - \widehat{m}\left(\mathbf{X}_i\right) - g\left(\mathbf{X}_i\right)^\top \left(h\left(\mathbf{T}_i\right) - \widehat{e}^h\left(\mathbf{X}_i\right)\right)\right)^2 + \Lambda\left(g\left(\cdot\right)\right) \right\} \tag{10}$$

## 4.2 Quasi-oracle error bound of Generalized Robinson Decomposition

We establish the main theoretical result of our paper: a *quasi-oracle convergence guarantee* for the Generalized Robinson Decomposition under a finite-basis representation of the outcome function. This result is analogous to the R-learner for binary CATEs [42]: when the true $e\left(\cdot\right), m\left(\cdot\right)$ are unknown, and we only have access to the estimators $\widehat{e}\left(\cdot\right), \widehat{m}\left(\cdot\right)$, then as long as the estimates converge at $n^{-1/4}$ rate, the estimator $\widehat{\tau}_\mathsf{b}\left(\cdot\right)$ achieves the same error bounds as an *oracle* who has ground-truth knowledge of these two nuisance components.

More formally, provided the nuisance estimators $\widehat{m}(\cdot)$ and $\widehat{e}^h(\cdot)$ converge at an $O\left(n^{-1/4}\right)$ rate, our CATE estimator will converge at an $\widetilde{O}(n^{-\frac{1}{2(1+p)}})$ rate for arbitrarily small $p > 0$, recovering the parametric convergence rate for when the true $m(\cdot)$ and $e^h(\cdot)$ are provided as oracle quantities.

Our analysis assumes that the outcome $\mathbb{E}\left[Y \mid \mathbf{X} = \mathbf{x}, \mathbf{T} = \mathbf{t}\right]$ can be written as a linear combination of fixed basis functions. By Proposition 1, as long as we have enough basis functions, this representation is flexible enough to capture the true outcome function.

**Assumption 4.** *Let $\boldsymbol{\alpha}(\mathbf{X}) \in \mathbb{R}^{d_{\boldsymbol{\alpha}}}$, $\boldsymbol{\beta}(\mathbf{T}) \in \mathbb{R}^{d_{\boldsymbol{\beta}}}$ be fixed, known orthonormal basis features on $\mathbf{X} \in \mathbb{R}^{d_{\mathbf{x}}}$, $\mathbf{T} \in \mathbb{R}^{d_{\mathbf{t}}}$, respectively. The true outcome function $f^*(\mathbf{x}, \mathbf{t}) = \mathbb{E}[Y \mid \mathbf{X} = \mathbf{x}, \mathbf{T} = \mathbf{t}]$ can be written as $f^*(\mathbf{x}, \mathbf{t}) = \boldsymbol{\alpha}^\top(\mathbf{x})\boldsymbol{\Theta}^*\boldsymbol{\beta}(\mathbf{t})$ for some (unknown) matrix of coefficients $\boldsymbol{\Theta}^*$.*

Note that by setting $g = \boldsymbol{\alpha}^\top\boldsymbol{\Theta}^*$ and $h = \boldsymbol{\beta}$, we recover eq. (7). Additionally, we will need overlap in the basis features $\boldsymbol{\alpha}(\mathcal{X}), \boldsymbol{\beta}(\mathcal{T})$.

**Assumption 5** (Overlap in features)**.** *The marginal distribution of features $\mathcal{P}_{\boldsymbol{\alpha}(\mathcal{X}) \times \boldsymbol{\beta}(\mathcal{T})}$ is positive, i.e. $\mathrm{supp}[\mathcal{P}_{\boldsymbol{\alpha}(\mathcal{X}) \times \boldsymbol{\beta}(\mathcal{T})}] = \boldsymbol{\alpha}(\mathcal{X}) \times \boldsymbol{\beta}(\mathcal{T})$.*

Assumption 5 is typically weaker than requiring overlap in $\mathbf{X}$ and $\mathbf{T}$, i.e., when $d_{\boldsymbol{\alpha}}, d_{\boldsymbol{\beta}} \ll d_{\mathbf{x}}, d_{\mathbf{t}}$.

With further technical assumptions specified in Appendix F, we establish the following theorem.

**Theorem 2.** *Let $\boldsymbol{\Theta}^*$ denote the representer of the true outcome function. Suppose Assumptions 5, 6, and 4 hold. Moreover, suppose that the propensity estimate $\widehat{e}^h$ is uniformly consistent,*

$$\sup_{\mathbf{x} \in \mathcal{X}} \|\widehat{e}^h(\mathbf{x}) - e^h(\mathbf{x})\| \to_p 0 \tag{11}$$

*and the $L_2$ errors converge at rate*

$$\mathbb{E}\left[\left\{\widehat{m}(\mathbf{X}) - m^*(\mathbf{X})\right\}^2\right], \mathbb{E}\left[\|\widehat{e}^h\left(\mathbf{X}\right) - e^h\left(\mathbf{X}\right)\|^2\right] = \mathcal{O}(a_n^2) \tag{12}$$

*for some sequence $a_n \to 0$, where $(a_n)$ is such that $a_n = O(n^{-\kappa})$ with $\kappa > \frac{1}{4}$. Further, we define the regret as the excess risk*

$$R\left(\widehat{\boldsymbol{\Theta}}_n\right) \triangleq L\left(\widehat{\boldsymbol{\Theta}}_n\right) - L\left(\boldsymbol{\Theta}^*\right), \; L\left(\boldsymbol{\Theta}\right) \triangleq \mathbb{E}\left[\left\{\left(Y - m^*\left(\mathbf{X}\right)\right) - \boldsymbol{\alpha}\left(\mathbf{X}\right)\boldsymbol{\Theta}\left(\boldsymbol{\beta}\left(\mathbf{T}\right) - e^h\left(\mathbf{X}\right)\right)\right\}^2\right]. \tag{13}$$

*Suppose that we obtain $\widehat{\boldsymbol{\Theta}}_n$ via a penalized basis function regression variant of the Generalized Robinson Decomposition, with a properly chosen penalty $\Lambda_n\left(\|\widehat{\boldsymbol{\Theta}}_n\|_2\right)$ (specified in the proof). Then, $\widehat{\boldsymbol{\Theta}}_n$ satisfies the regret bound: $R\left(\widehat{\boldsymbol{\Theta}}_n\right) = \widetilde{O}(r_n^2)$ with $r_n = n^{-\frac{1}{2(1+p)}}$ for arbitrarily small $p > 0$.*

## 5 Structured Intervention Networks

We introduce *Structured Intervention Networks* (SIN), a two-stage training algorithm for neural networks, which enables flexibility in learning complex causal relationships, and scalability to large data-sets. This implementation of GRD strikes a balance between theory and practice: while we assumed fixed basis-functions in Section 4.2, in practice, we often need to learn the feature maps from data. We leave the convergence analysis of this representation learning setting for future work.

### 5.1 Training Algorithm

We propose to simultaneously learn feature maps $\widehat{g}\left(\mathbf{X}\right), \widehat{h}\left(\mathbf{T}\right)$ using alternating gradient descent, so that they can adapt to each other. A remaining challenge is that learning $\widehat{e}^h\left(\mathbf{X}\right)$ is now entangled with learning $\widehat{h}\left(\mathbf{T}\right)$. While the R-learner is based on the idea of *cross-fitting*, where at each data point $i$ we pick estimates of the nuisances that do not use that data point, we introduce a pragmatic representation learning approach for $(\widehat{g}, \widehat{h})$ that does not use cross-fitting[3].

---

[3]We could in principle use cross-fitting for $\widehat{e}^h$, although the loop between fitting $\widehat{h}$ alternating with $\widehat{e}^h$ would break the overall independence between $\widehat{e}_i^h(\mathbf{X})$ and data point $i$. While it is possible that cross-fitting for $\widehat{e}^h$ is still beneficial in this case, for simplicity and for computational savings, we did not implement it.

**a** SIN Training.

**Input**: Stage 1 data $\mathcal{D}_1 := \{(\mathbf{x}_i, y_i)\}_{i=1}^m$, Stage 2 data $\mathcal{D}_2 := \{(\mathbf{x}_i, \mathbf{t}_i, y_i)\}_{i=1}^n$ Step sizes $\lambda_{\boldsymbol{\theta}}, \lambda_{\boldsymbol{\eta}}, \lambda_{\boldsymbol{\psi}}, \lambda_{\boldsymbol{\phi}}$. Number of update steps $K$. Mini-batch sizes $B_1, B_2$.

1: Initialize parameters: $\boldsymbol{\theta}, \boldsymbol{\eta}, \boldsymbol{\psi}, \boldsymbol{\phi}$
2: **while** not converged **do**          ▷ *Stage 1*
3:     Sample mini-batch $\{(\mathbf{x}_b, y_b)\}_{b=1}^{m_{B_1}}$
4:     Evaluate $J_m(\boldsymbol{\theta})$
5:     Update $\boldsymbol{\theta} \leftarrow \boldsymbol{\theta} - \lambda_{\boldsymbol{\theta}} \widehat{\nabla}_{\boldsymbol{\theta}} J(\boldsymbol{\theta})$
6: **end while**
7: **while** not converged **do**          ▷ *Stage 2*
8:     Sample mini-batch $\{(\mathbf{x}_b, \mathbf{t}_b, y_b)\}_{b=1}^{n_{B_2}}$
9:     Evaluate $J_{g,h}(\boldsymbol{\psi}, \boldsymbol{\phi}), J_{e^h}(\boldsymbol{\eta})$
10:    **for** $k = 1$ to $K$ **do**
11:        Update $\boldsymbol{\phi} \leftarrow \boldsymbol{\phi} - \lambda_{\boldsymbol{\phi}} \widehat{\nabla}_{\boldsymbol{\phi}} J_{g,h}(\boldsymbol{\psi}, \boldsymbol{\phi})$
12:        Update $\boldsymbol{\psi} \leftarrow \boldsymbol{\psi} - \lambda_{\boldsymbol{\psi}} \widehat{\nabla}_{\boldsymbol{\psi}} J_{g,h}(\boldsymbol{\psi}, \boldsymbol{\phi})$
13:    **end for**
14:    Update $\boldsymbol{\eta} \leftarrow \boldsymbol{\eta} - \lambda_{\boldsymbol{\eta}} \widehat{\nabla}_{\boldsymbol{\eta}} J_{e^h}(\boldsymbol{\eta})$
15: **end while**

**b** Pseudocode in a PyTorch-like style.

```
# Initialize submodels and optimizers
m, e, g, h = MLP(...), MLP(...), MLP(...),
    GNN(...)
m_opt, e_opt, g_opt, h_opt = Adam(m.params(),
    m_lr), Adam(e.params(), e_lr), ...

# Stage 1
for batch in train_loader:
    X, Y = batch.X, batch.Y
    m_opt.zero_grad()
    F.mse_loss(m(X), Y).backward()
    m_opt.step()

# Stage 2
for batch in train_loader:
    X, T, Y = batch.X, batch.T, batch.Y
    for _ in range(num_update_steps):
        g_opt.zero_grad()
        h_opt.zero_grad()
        F.mse_loss((g(X)*(h(T) - e(X))).sum
            (-1), (Y-m(X))).backward()
        g_opt.step()
        h_opt.step()
    e_opt.zero_grad()
    F.mse_loss(e(X), h(T)).backward()
    e_opt.step()
```

Figure 2: The two-stage algorithm for training SIN.

We learn surrogate models for the mean outcome and propensity features $\widehat{m}_{\boldsymbol{\theta}}(\mathbf{X})$ and $\widehat{e}_{\boldsymbol{\eta}}^h(\mathbf{X})$ with parameters $\boldsymbol{\theta} \in \mathbb{R}^{d_{\boldsymbol{\theta}}}, \boldsymbol{\eta} \in \mathbb{R}^{d_{\boldsymbol{\eta}}}$, as well as feature maps for covariates and treatments $\widehat{g}_{\boldsymbol{\psi}}(\mathbf{X}), \widehat{h}_{\boldsymbol{\phi}}(\mathbf{T})$, parameterized by $\boldsymbol{\psi} \in \mathbb{R}^{d_{\boldsymbol{\psi}}}, \boldsymbol{\phi} \in \mathbb{R}^{d_{\boldsymbol{\phi}}}$. We denote regularizers by $\Lambda(\cdot)$. Figure 2 summarizes the algorithm. As the mean outcome model $\widehat{m}_{\boldsymbol{\theta}}(\mathbf{X})$ does not depend on the other components, we learn it separately in Stage 1. In Stage 2, we alternate between learning $\boldsymbol{\psi}, \boldsymbol{\phi}, \boldsymbol{\eta}$.

**Stage 1:**   Learn parameters $\boldsymbol{\theta}$ of the mean outcome model $\widehat{m}_{\boldsymbol{\theta}}(\mathbf{X})$ based on the objective

$$ J_m(\boldsymbol{\theta}) = \frac{1}{m} \sum_{i=1}^m \left( y_i - \widehat{m}_{\boldsymbol{\theta}}(\mathbf{x}_i) \right)^2 + \Lambda(\boldsymbol{\theta}), \tag{14} $$

which relies only on covariates and outcome data $\mathcal{D}_1 := \{(\mathbf{x}_i, y_i)\}_{i=1}^m$.

**Stage 2:**   Learn parameters $\boldsymbol{\psi}, \boldsymbol{\phi}$ for the covariates and treatments feature maps $\widehat{g}_{\boldsymbol{\psi}}(\mathbf{X}), \widehat{h}_{\boldsymbol{\phi}}(\mathbf{T})$, as well as parameters $\boldsymbol{\eta}$ for the propensity features $\widehat{e}_{\boldsymbol{\eta}}^h(\mathbf{X})$.

$$ J_{g,h}(\boldsymbol{\phi}, \boldsymbol{\psi}) = \frac{1}{n} \sum_{i=1}^n \left( y_i - \left\{ \widehat{m}_{\boldsymbol{\theta}}(\mathbf{x}_i) + \widehat{g}_{\boldsymbol{\psi}}(\mathbf{x}_i)^\top \left( \widehat{h}_{\boldsymbol{\phi}}(\mathbf{t}_i) - \widehat{e}_{\boldsymbol{\eta}}^h(\mathbf{x}_i) \right) \right\} \right)^2 + \Lambda(\boldsymbol{\psi}) + \Lambda(\boldsymbol{\phi}). \tag{15} $$

This loss hinges on $\widehat{e}_{\boldsymbol{\eta}}^h(\mathbf{X})$, which needs to be learned by

$$ J_{e^h}(\boldsymbol{\eta}) = \sum_{i=1}^n \left\| \widehat{h}_{\boldsymbol{\phi}}(\mathbf{t}_i) - \widehat{e}_{\boldsymbol{\eta}}^h(\mathbf{x}_i) \right\|_2^2 + \Lambda(\boldsymbol{\eta}), \tag{16} $$

note again the dependence on $\widehat{h}_{\boldsymbol{\phi}}(\mathbf{T})$. While it may be tempting to learn $\boldsymbol{\psi}, \boldsymbol{\phi}$ and $\boldsymbol{\eta}$ jointly, they have fundamentally different objectives ($\widehat{e}_{\boldsymbol{\eta}}^h(\mathbf{X})$ is defined as an estimate of the expectation $\mathbb{E}[h(\mathbf{T}) \mid \mathbf{x}]$). Therefore, we employ an alternating optimization procedure, where we take $k \in \{1, \ldots, K\}$ optimization steps for $\boldsymbol{\psi}, \boldsymbol{\phi}$ towards $J_{g,h}(\boldsymbol{\psi}, \boldsymbol{\phi})$ and one step for learning $\boldsymbol{\eta}$. We observe that setting $K > 1$, i.e. updating $\boldsymbol{\psi}, \boldsymbol{\phi}$ more frequently than $\boldsymbol{\eta}$, stabilizes the training process.

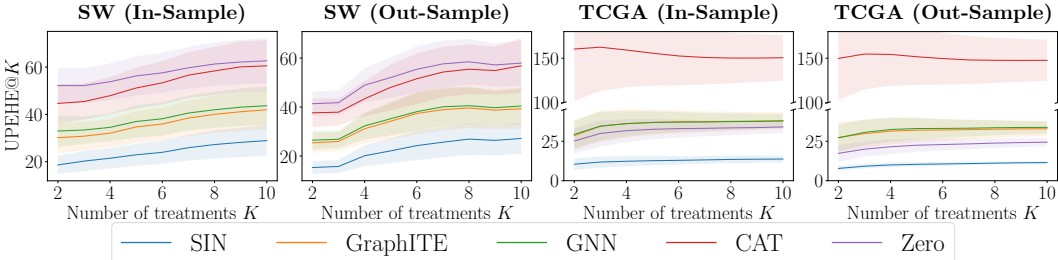

Figure 3: UPEHE@$K$ for $K \in \{2, \ldots, 10\}$.

## 5.2 Advantages of SIN

We conclude by describing the beneficial properties of SIN, particularly in finite-sample regimes:

1. **Targeted regularization**: Regularizing $\widehat{g}_\psi(\mathbf{X}), \widehat{h}_\phi(\mathbf{T})$ in eq. (15) after partialing out confounding is a type of targeted regularization of the isolated causal effect. In contrast, outcome estimation methods can suffer from regularization-induced confounding, e.g., regularizing the effect estimate away from zero in the service of trying to improve predictive performance [29].

2. **Propensity features**: Learning propensity features can help us to (i) partial out parts of $\mathbf{X}$ that cause the treatment but not the outcome, and (ii) dispose unnecessary components of $\mathbf{T}$.

3. **Data-efficiency**: In contrast to methods that split the data into disjoint models for each treatment group (known as *T-learners* for binary treatments [8, 10]), sharing causal effect parameters between all covariates regardless of their assigned treatment increases data-efficiency.

4. **Partial data**: In settings without access to both the treatment assignment and the outcome but only access to one of them, one can leverage that data to improve the (nuisance) estimator further, e.g., when a patient's recovery is observed one year after a drug was administered [33].

## 6 Experiments

Here we evaluate how CATE estimation with our proposed model SIN compares with prior methods.

### 6.1 Experimental Setup

**Datasets.** To be able to compute CATE estimation error w.r.t. a ground truth, we design two causal models: a simpler synthetic model with small-world graph treatments and a more complex model with real-world molecular graph treatments and gene expression covariates. The Small-World (SW) simulation contains 1,000 uniformly sampled covariates and 200 randomly generated Watts–Strogatz small-world graphs [61] as treatments. *The Cancer Genomic Atlas* (TCGA) simulation uses 9,659 gene expression measurements of cancer patients for covariates [62] and 10,000 sampled molecules from the QM9 dataset [46] as treatments. Appendix D details the data-generating schemes.

**Baselines.** We compare our method to (1) **Zero**, a sanity-check baseline that consistently predicts zero treatment effect and equals the mean squared treatment effect (poorly regularized models may perform worse than that due to confounding), (2) **CAT**, a categorical treatment variable model using one-hot encoded treatment indicator vectors, (3) **GNN**, a model that first encodes treatments with a GNN and then concatenates treatment and individual features for regression, (4) **GraphITE** [16], a CATE estimation method designed for graph treatments (more details in Section 2). GNN and CAT reflect the performance of standard regression models. The contrast between these two provides insight into whether the additional graph structure of the treatment improves CATE estimation. To deal with unseen treatments during CATE evaluation, we map such to the most similar ones seen during training based on their Euclidean distance in the embedding space of the GNN baseline.

**Graph models.** For small-world networks, we use *k-dimensional GNNs* [38], as to distinguish graphs they take higher-order structures into account. To model molecular graphs, we use *Relational Graph Convolutional Networks* [50], where the nodes are atoms and each edge type corresponds to a specific bond type. We use the implementations of PyTorch Geometric [11].

Table 1: Error of CATE estimation for all methods, measured by WPEHE@6. Results are averaged over 10 trials, $\pm$ denotes std. error (each trial samples treatment assignment matrix $\mathbf{W}$).

| Method | SW | | TCGA | |
|---|---|---|---|---|
| | In-sample | Out-sample | In-sample | Out-sample |
| Zero | $56.26 \pm 8.12$ | $53.77 \pm 8.93$ | $26.63 \pm 7.55$ | $17.94 \pm 4.86$ |
| CAT | $51.75 \pm 8.85$ | $49.76 \pm 9.73$ | $155.88 \pm 52.82$ | $146.62 \pm 42.32$ |
| GNN | $37.10 \pm 6.84$ | $36.74 \pm 7.42$ | $30.67 \pm 8.29$ | $27.57 \pm 7.95$ |
| GraphITE | $34.81 \pm 6.70$ | $35.94 \pm 8.07$ | $30.31 \pm 8.96$ | $27.48 \pm 8.95$ |
| **SIN** | $\mathbf{23.00 \pm 4.56}$ | $\mathbf{23.19 \pm 5.56}$ | $\mathbf{10.98 \pm 3.45}$ | $\mathbf{8.15 \pm 1.46}$ |

**Evaluation metrics.** We extend the *expected Precision in Estimation of Heterogeneous Effect* (PEHE) commonly used in binary treatment settings [19] to arbitrary pairs of treatments $(\mathbf{t}, \mathbf{t}')$ as follows. We denote the *Unweighted PEHE* (UPEHE) and the *Weighted PEHE* (WPEHE) as

$$\epsilon_{\text{UPEHE(WPEHE)}} \triangleq \int_{\mathcal{X}} \left( \widehat{\tau}\left(\mathbf{t}', \mathbf{t}, \mathbf{x}\right) - \tau\left(\mathbf{t}', \mathbf{t}, \mathbf{x}\right) \right)^2 p\left(\mathbf{t} \mid \mathbf{x}\right) p\left(\mathbf{t}' \mid \mathbf{x}\right) p(\mathbf{x}) \, d\mathbf{x}, \qquad (17)$$

where the weighted version gives less importance to treatment pairs that are less likely; to account for the fact that such pairs will have higher estimation errors. In fact, as the reliability of estimated effects decreases by how likely they are in the observational study, we evaluate all methods on U/WPEHE truncated to the top $K$ treatments, which we call U/WPEHE@$K$. To compute this, for each $\mathbf{x}$, we rank all treatments by their propensity $p\left(\mathbf{t} \mid \mathbf{x}\right)$ (given by the causal model) in descending order. We take the top $K$ treatments and compute the U/WPEHE for all $\binom{K}{2}$ treatment pairs.

**In-sample vs. out-sample.** A common benchmark for causal inference methods is the *in-sample* task, which we include here for completeness: estimating CATEs for covariate values $\mathbf{x}$ found in the training set. This task is still non-trivial, as the outcome of only one treatment is observed during training [4]. In contrast, and arguably of more relevance to decision making, the goal of the *out-sample* task is to estimate CATEs for completely unseen covariate realizations $\mathbf{x}'$.

**Hyper-parameter tuning.** To ensure a fair comparison, we perform hyper-parameter optimization with random search for all models on held-out data and select the best hyper-parameters over 10 runs.

**Propensity.** We define the propensity (or *treatment selection bias*) as $p\left(\mathbf{T} \mid \mathbf{x}\right) = \text{softmax}\left(\kappa \mathbf{W}^{\top} \mathbf{X}\right)$, where $\mathbf{W} \in \mathbb{R}^{|\mathcal{T}| \times d}, \forall i, j : W_{ij} \sim \mathcal{U}\left[0, 1\right]$ is a random matrix (sampled then fixed for each run). Recall $|\mathcal{T}|$ is the number of available treatments and let $d$ be the dimensionality of the covariates. Here the *bias strength* $\kappa$ is a temperature parameter that determines the flatness of the propensity (the lower the flatter, i.e., $\kappa = 0$ corresponds to the uniform distribution).

### 6.2 Comparison of Performances on different $K$ Treatments

Figure 3 shows the UPEHE@$K$ of all methods for $K \in \{2, \ldots, 10\}$. We also report the WPEHE@6 of all methods in Table 1. Unless stated otherwise, we report results for bias strengths $\kappa = 10$ and $\kappa = 0.1$ in the SW and TCGA datasets, respectively across 10 random trials.

The results indicate that the relative performance of each method, for both the in-sample and out-sample estimation tasks, is consistent. Further, they suggest that, overall, the performance of SIN is best due to a better isolation of the causal effect from the observed data compared to other methods. The performance difference between CAT and GNN across all results indicate that accounting for graph information significantly improves the estimates. We observe from the SW experiments that GraphITE [16] performs slightly better than GNN, while it is nearly the same as GNN on TCGA.

Surprisingly, the results of the TCGA experiments with low bias strength $\kappa = 0.1$ expose that all models but SIN fail to isolate causal effects better than the Zero baseline. These results confirm that confounding effects of $\mathbf{X}$ on $Y$ combined with moderate causal effects can cause severe regularization bias for black-box regression models, while SIN partials these out from the outcome by $\widehat{m}_{\boldsymbol{\theta}}\left(\mathbf{X}\right)$. We include additional results on convergence and larger values of $K$ in Appendix E.1.

---

[4]The original motivation comes from Fisherian designs where the only source of randomness is on the treatment assignment [20]. Our motivation is simpler: rule out the extra variability from different covariates, highlighting the difference between methods due to different loss functions and less due to smoothing abilities.

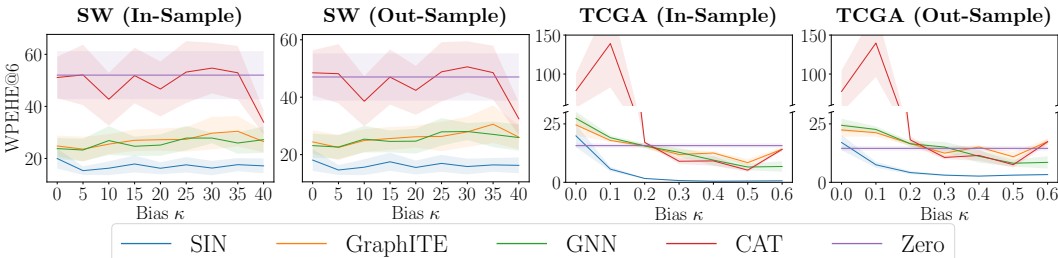

Figure 4: WPEHE@6 over increasing bias strength $\kappa$.

## 6.3 Comparison of Robustness to different Bias Strengths $\kappa$

A strong selection bias (i.e. large $\kappa$) in the observed data makes CATE estimation more difficult, as it becomes unlikely to see certain treatments $\mathbf{t} \in \mathcal{T}$ for particular covariates $\mathbf{x}$. Here, we assess each model's robustness to varying levels of selection bias, determined by $\kappa$, across 5 random seeds. In Figure 4, we see that SIN outperforms the baselines across the entire range of considered biases. Interestingly, SIN performs competitively even in a case with no selection bias ($\kappa = 0$, which corresponds to a randomized experiment). Importantly, all performances seem to either stagnate (SW) or to increase (TCGA) with increasing biases. Notably, the poor performance of CAT suddenly improves on datasets with high bias. We believe this is because, in high bias regimes, we see fewer distinct treatments overall, which allows the CAT model to approach the performance of GNN.

## 7 Limitations, Future Work and Potential Negative Societal Impacts

**Limitations and future work.**   Firstly, in some real-life domains, Assumption 1 (Unconfoundedness) can be too strong, as there may exist *hidden confounders*. There are two common strategies to deal with them: utilizing *instrumental variables* [17, 58, 63] or *proxy variables* [35, 37, 59]. Developing new approaches for structured interventions in such settings is a promising future direction. Secondly, SIN is based on neural networks; however, neural network initialization can impact final estimates. To obtain consistency guarantees, GRD can be combined with kernel methods [35, 58].

**Potential negative societal impacts.**   Because causal inference methods make recommendations about interventions to apply in real-world settings, misapplying them can have a negative real-world impact. It is crucial to thoroughly test these methods on realistic simulations and alter aspects of them to understand how violations of assumptions impact estimation. We have aimed to provide a comprehensive evaluation of structured treatment methods by showing how estimation degrades as less likely treatments are considered (Figure 3) and as treatment bias increases (Figure 4).

## 8 Conclusion

The main contributions of this paper are two-fold: (i) the generalized Robinson decomposition that yields a pseudo-outcome targeting the causal effect while possessing a quasi-oracle convergence guarantee under mild assumptions, and (ii) Structured Intervention Networks, a practical algorithm using representation learning that outperforms prior approaches in experiments with graph treatments.

## Acknowledgements

We thank Antonin Schrab, David Watson, Jakob Zeitler, Limor Gultchin, Marc Deisenroth and Shonosuke Harada for useful discussions and constructive feedback on the paper. JK and YZ acknowledge support by the Engineering and Physical Sciences Research Council with grant number EP/S021566/1. This work was partially supported by an ONR grant number N62909-19-1-2096 to RS. We thank the Alan Turing Institute for the provision of Azure cloud computing resources.

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
