Define *propensity features* $e^h(\mathbf{x}) \triangleq \mathbb{E}[h(\mathbf{T}) \mid \mathbf{x}]$ and $m(\mathbf{x}) \triangleq \mathbb{E}[Y \mid \mathbf{x}] = g(\mathbf{x})^\top e^h(\mathbf{x})$.

Following the same steps as in Section 3.2, the Generalized Robinson Decomposition for eq. (7) is

$$\boxed{Y - m(\mathbf{X}) = g(\mathbf{X})^\top \left( h(\mathbf{T}) - e^h(\mathbf{X}) \right) + \varepsilon.} \tag{9}$$

Given nuisance estimates $\widehat{m}(\cdot), \widehat{e}^h(\cdot)$, we can use this decomposition to derive an optimization problem for $h(\cdot), g(\cdot)$ (note $\widehat{e}^h(\cdot)$ implicitly depends on $h(\cdot)$, we address this dependence in Section 5).

$$\widehat{g}(\cdot), \widehat{h}(\cdot) \triangleq \arg\min_{g,h} \left\{ \frac{1}{n} \sum_{i=1}^{n} \left( Y_i - \widehat{m}(\mathbf{X}_i) - g(\mathbf{X}_i)^\top \left( h(\mathbf{T}_i) - \widehat{e}^h(\mathbf{X}_i) \right) \right)^2 + \Lambda(g(\cdot)) \

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

## A   Other Related Work

**Plug-in estimators.**   A recent line of work for CATE estimation derives *plug-in estimators* [8].[5] These work by decomposing CATE estimation into multiple sub-problems (so-called *nuisance components*), each solvable using any supervised learning method [10, 15, 26, 29, 42]. Currently, these approaches are limited to binary treatment setups. Our approach is inspired by these methods, extending plug-in estimation to structured treatment settings.

---

[5]These are also called *meta-learners*. To avoid confusion with *meta-learning*, we call these *plug-in estimators*.

**CATE estimation with neural networks.** Neural network CATE estimators typically use separate prediction heads for each treatment option [24, 32, 41, 51, 52, 54, 55]. This architectural design reduces one source of regularization bias: the influence of the treatment indicator variable might be lost in the high-dimensional network representations. Extending this idea directly to structured treatments would not only be computationally expensive, but would also not be able to make use of treatment features or learn treatment representations.

**Multiple treatments.** While Inverse Probability Weighting (IPW) [30, 31, 66] is a popular technique for estimating effects with multiple, categorical treatments, it requires estimating the propensity density which is infeasible in settings with hundreds or thousands of treatments; some of which may have not been seen during training. Nabi et al. [39] propose a framework for sufficient dimensionality reduction of high-dimensional treatments based on semiparametric inference theory. Besides relying on IPW, this approach is designed for average treatment effects (not CATEs).

# B  The Generalized Robinson Decomposition

## B.1  Motivation

frequently the influence of $\mathbf{T}$ on $Y$ is very different from the influence of $\mathbf{X}$ on $Y$. Specifically, $f(\mathbf{X}, \mathbf{T})$ often has different smoothness in $\mathbf{X}$ and $\mathbf{T}$. For instance, different health histories $\mathbf{X}$ for a fixed treatment $\mathbf{t}$ will have a much more variable effect on $Y$ than different treatments $\mathbf{t}$ for a given history $\mathbf{X}$. This is why methods like the R-learner [42] have carefully separated estimation functions of $\mathbf{X}$ from functions of $\mathbf{T}$ [8].

A generic way to extend the Robinson decomposition to arbitrary treatments is to learn a model $\hat{f}(\mathbf{X}, \mathbf{T})$ defined over the entire outcome surface, via mean outcome $\hat{m}(\mathbf{X})$ and treatment conditional density $p(\mathbf{T} \mid \mathbf{X})$. In this case, we fit the relationship

$$Y - m(\mathbf{X}) = f(\mathbf{X}, \mathbf{T}) - e^p(\mathbf{X}) + \varepsilon, \quad \text{where} \quad e^p(\mathbf{x}) \triangleq \mathbb{E}\left[f(\mathbf{X}, \mathbf{T}) \mid \mathbf{x}\right]. \tag{18}$$

To learn $f(\cdot, \cdot)$ from a dataset $\mathcal{D} = \left\{(\mathbf{x}_i, \mathbf{t}_i, y_i)\right\}_{i=1}^n$ we need to solve,

$$\hat{f} = \underset{f \in \mathcal{F}}{\arg\min} \frac{1}{N} \sum_{i=1}^N \left[\left\{y_i - \hat{m}(\mathbf{x}_i)\right\} - \left(f(\mathbf{x}_i, \mathbf{t}_i) - \hat{e}^p(\mathbf{x}_i)\right)\right]^2, \tag{19}$$

where $\mathcal{F}$ is some function space and $\hat{m}$ is a plug-in finite sample estimate of $m$. Because $e^p$ contains $f$, we need to estimate it, which we denote $\hat{e}^p$.

One solution is to estimate the propensity $p(\mathbf{T} \mid \mathbf{X})$ and use it to compute $\hat{e}^p$. However, this approach requires conditional density estimation over potentially high-dimensional, structured treatments, which remains an open research question [65], and is prone to high variance [58]. Further, to compute $\hat{e}^p$ from it, one has to resort to Monte Carlo evaluation. By learning propensity features $\hat{e}_{\boldsymbol{\eta}}^h(\mathbf{X})$ instead of $p(\mathbf{T} \mid \mathbf{X})$, we avoid these issues.

Another option is to solve for $f$, fix it, then estimate $\hat{e}^p$ using regression from finite samples, and iterate to a fixed point. However, there is a fundamental issue with this approach: we are typically interested in regularizing the causal effect directly as opposed to the generic regression function. This is why, for instance, the R-learner parameterizes $\mu_1(\mathbf{x})$ as a (nuisance) baseline $\mu_0(\mathbf{x})$ plus the CATE $\tau_\mathrm{b}(\mathbf{x})$. The black-box $f(\mathbf{x}, \mathbf{t})$ does not capture the asymmetry between $\mathbf{x}$ and $\mathbf{t}$ in the implied CATE $f(\mathbf{x}, \mathbf{t}) - f(\mathbf{x}, \mathbf{t}')$. Further, unlike the binary case, in many applications, we do not have a baseline treatment $\mathbf{t}_0$ with respect to which we could parameterize $f$ in terms of some $\tau(\mathbf{t}, \mathbf{t}_0, \mathbf{x})$. To regularize the causal effect more directly, we make the product effect assumption, which allows us to partial out confounding.

## B.2  Derivation in detail

We consider the product effect parameterization of $p(y \mid \mathbf{x}, \mathbf{t})$,

$$Y = \underbrace{g(\mathbf{X})^\top h(\mathbf{T})}_{=:f(\mathbf{X},\mathbf{T})} + \varepsilon, \tag{20}$$

where $g : \mathcal{X} \to \mathbb{R}^d, h : \mathcal{T} \to \mathbb{R}^d$ and $\mathbb{E}[\varepsilon \mid \mathbf{x}, \mathbf{t}] = \mathbb{E}[\varepsilon \mid \mathbf{x}] = 0$, for all $(\mathbf{x}, \mathbf{t}) \in \mathcal{X} \times \mathcal{T}$. Rearranging eq. (20) yields the Robinson residual

$$\varepsilon = Y - g(\mathbf{X})^\top h(\mathbf{T}), \tag{21}$$

which we aim to rewrite in terms of $m(\mathbf{X})$. To this end, we define *propensity features* $e^h(\mathbf{X})$ as

$$e^h(\mathbf{X}) \triangleq \mathbb{E}\left[h(\mathbf{T}) \mid \mathbf{X}\right], \quad \text{such that} \quad m(\mathbf{X}) = \mathbb{E}\left[Y \mid \mathbf{X}\right] = g(\mathbf{X})^\top e^h(\mathbf{X}). \tag{22}$$

To obtain the generalized Robinson decomposition, one rewrites eq. (21) as

$$\varepsilon = Y - \left(g(\mathbf{X})^\top \left[h(\mathbf{T}) + e^h(\cancel{\mathbf{X}}) - e^h(\cancel{\mathbf{X}})\right]\right) \tag{23}$$

$$= Y - \left(g(\mathbf{X})^\top e^h(\mathbf{X}) + g(\mathbf{X})^\top \left(h(\mathbf{T}) - e^h(\mathbf{X})\right)\right) \tag{24}$$

$$= Y - \underbrace{\left(g(\mathbf{X})^\top e^h(\mathbf{X})\right)}_{m(\mathbf{X})} - g(\mathbf{X})^\top \left(h(\mathbf{T}) - e^h(\mathbf{X})\right). \tag{25}$$

Hence, the generalized Robinson decomposition is

$$Y - m(\mathbf{X}) = g(\mathbf{X})^\top \left(h(\mathbf{T}) - e^h(\mathbf{X})\right) + \varepsilon. \tag{26}$$

## C  Universality of Product Decomposition

**Proof of Proposition 1.**

*Proof.* Define $\mathcal{H}_{0\mathcal{X} \times \mathcal{T}}$ as $\left\{ f(\mathbf{x}, \mathbf{t}) = \sum_{i=1}^n \alpha_i k\left((\mathbf{x}_i, \mathbf{t}_i), (\mathbf{x}, \mathbf{t})\right) \mid n \in \mathbb{N}, \alpha_{i=1,\cdots,n} \in \mathbb{R} \right\}$. By definition, the RKHS $\mathcal{H}_{\mathcal{X} \times \mathcal{T}}$ is the set of pointwise limits of Cauchy sequences $(f_n)_n \in \mathcal{H}_{0\mathcal{X} \times \mathcal{T}}$. By Lemma 41 of [53], the Cauchy sequences also converges in the $\mathcal{H}_{\mathcal{X} \times \mathcal{T}}$ norm.

For any $f \in \mathcal{H}_{\mathcal{X} \times \mathcal{T}}$, pick its Cauchy sequence $(f_n) in \mathcal{H}_{0\mathcal{X} \times \mathcal{T}}$. Since $\sum_{i=1}^\infty \alpha_i k\left((\mathbf{x}_i, \mathbf{t}_i), (\mathbf{x}, \mathbf{t})\right)$ converges in $\|\cdot\|_{\mathcal{H}_{\mathcal{X} \times \mathcal{T}}}$, for any $\tilde{\epsilon}$ there exist a $\tilde{d}$ such that let $f_{\tilde{d}} = \sum_{i=1}^{\tilde{d}} \alpha_i k\left((\mathbf{x}_i, \mathbf{t}_i), (\mathbf{x}, \mathbf{t})\right)$, then

$$\|f_{\tilde{d}} - f\|_{\mathcal{H}_{\mathcal{X} \times \mathcal{T}}} \leq \tilde{\epsilon} \tag{27}$$

Since for any RKHS with kernel $k$, the RKHS norm is always an upper bound on the $L_2$ norm up to scaling by a constant $C_k$,

$$\|f_{\tilde{d}} - f\|_{L_2(P_{(\mathcal{X} \times \mathcal{T})})} \leq C_k \|f_{\tilde{d}} - f\|_{\mathcal{H}_{\mathcal{X} \times \mathcal{T}}} \leq C_k \tilde{\epsilon} \tag{28}$$

Then for any $\epsilon$, we can choose $d \in \mathbb{N}$ such that $\|f_d - f\|_{L_2(P_{\mathcal{X} \times \mathcal{T}})} \leq C_k \cdot \frac{\epsilon}{C_k} = \epsilon$.

It remains to show that $f_d$ can be written as $g^\top h$ as required. $\mathcal{H}_{\mathcal{X} \times \mathcal{T}}$ is isometrically isomorphic to $\mathcal{H}_{\mathcal{X}} \times \mathcal{H}_{\mathcal{T}}$; we can decompose $k$ into the product kernel

$$k\left((\mathbf{x}, \mathbf{t}), (\mathbf{x}', \mathbf{t}')\right) = k_{\mathcal{X}}(\mathbf{x}, \mathbf{x}') k_{\mathcal{T}}(\mathbf{t}, \mathbf{t}'). \tag{29}$$

Thus $f_d(\mathbf{x}, \mathbf{t}) = \sum_{i=1}^d \alpha_i k_{\mathcal{X}}(\mathbf{x}, \mathbf{x}_i) k_{\mathcal{T}}(\mathbf{t}, \mathbf{t}_i)$. Set $g(\mathbf{x}) = \left(\alpha_1 k_{\mathcal{X}}(\mathbf{x}, \mathbf{x}_1), \cdots, \alpha_d k_{\mathcal{X}}(\mathbf{x}, \mathbf{x}_d)\right)^\top$, $h(\mathbf{t}) = \left(k_{\mathcal{T}}(\mathbf{t}, \mathbf{t}_1), \cdots, k_{\mathcal{T}}(\mathbf{t}, \mathbf{t}_d)\right)^\top$, we obtain $f_d = g^\top$. $\qquad \square$

## D  Experimental Details

### D.1  Simulations

**Baseline effect**  Similarly as in [7, 10, 41], for each run of the experiment, we randomly sample a vector $\mathbf{u}_0 \sim \mathcal{U}(\mathbf{0}, \mathbf{1})$, and set $\mathbf{v}_0 = \mathbf{u}_0 / \|\mathbf{u}_0\|$ where $\|\cdot\|$ is the Euclidean norm. We then model the baseline effect as

$$\mu_0(\mathbf{x}) = \mathbf{v}_0^\top \mathbf{x}. \tag{30}$$

### D.1.1 Small-World Networks

**Covariates**   We uniformly sample 20-dimensional multivariate covariates $\mathbf{X} \sim \mathcal{U}(-\mathbf{1}, \mathbf{1})$. The in-sample dataset consists of $1{,}000$ units, and the out-sample one of $500$. For the treatment assignment, we square the covariates element-wise; i.e., we sample treatment assignments according to $p\left(\mathbf{T} \mid \mathbf{x}^2\right)$.

**Graph interventions**   For each graph intervention, we uniformly sample a number of nodes between $10$ and $120$, number of neighbors for each node between $3$ and $8$, and the probability of rewiring each edge between $0.1$ and $1$ Then, we repeatedly generate Watts–Strogatz small-world graphs until we get a connected one. Each vertex has one feature, which is its degree centrality. We denote a graph's node connectivity as $\nu(\mathcal{G})$ and its average shortest path length as $l(\mathcal{G})$.

**Outcomes**   Analogously as for the baseline effect, we generate two randomly sampled vectors $\mathbf{v}_\nu$ and $\mathbf{v}_l$. Then, given an assigned graph treatment $\mathcal{G}$ and a covariate vector $\mathbf{x}$, we generate the outcome as

$$Y = 100\mu_0(\mathbf{x}) + 0.2\nu(\mathcal{G})^2 \cdot \mathbf{v}_\nu^\top \mathbf{x} + l(\mathcal{G}) \cdot \mathbf{v}_l^\top \mathbf{x} + \epsilon, \quad \epsilon \sim \mathcal{N}(0,1). \tag{31}$$

### D.1.2 TCGA

**Covariates**   The *The Cancer Genomic Atlas* (TCGA) simulation uses $4{,}000$-dimensional $9{,}659$ gene expression measurements of cancer patients for covariates [62], i.e., each unit is a covariate vector $\mathbf{X} \in \mathbb{R}^{4000}$. The in-sample and out-sample datasets consist of $5{,}000$ and $4{,}659$ units, respectively. In each run, the units are split randomly into in- and out-sample datasets. We used the same version of the TCGA dataset as used by Bica et al. [7] and Schwab et al. [52].

**Graph interventions**   In each run, we randomly sample $10{,}000$ molecules from the Quantum Machine 9 (QM9) dataset [46, 49] (with 133k molecules in total). For each molecule, we create a relational graph, where each node corresponds to an atom and consist of 78 atom features. An edge corresponds to the chemical bond type, where we label each edge correspondingly, considering *single*, *double*, *triple* and *aromatic* bonds. Furthermore, for each molecule, we obtain 8 of its properties *mu, alpha, homo, lumo, gap, r2, zpve, u0*, which we collect in the vector $\mathbf{z} \in \mathbb{R}^8$.

**Outcomes**   For each covariate vector $\mathbf{x}$, we compute its 8-dimensional PCA components, denoted by $\mathbf{x}^{(\text{PCA})} \in \mathbb{R}^8$. Then, given the molecular properties of the assigned molecule treatment $\mathbf{z}$, we generate outcomes by

$$Y = 10\mu_0(\mathbf{x}) + 0.01\mathbf{z}^\top \mathbf{x}^{(\text{PCA})} + \epsilon, \quad \epsilon \sim \mathcal{N}(0,1). \tag{32}$$

### D.2 Hyper-parameters

To ensure a fair comparison between all models, we perform hyper-parameter optimization with random search for all models on held-out data and select the best hyper-parameters over 10 runs. While conceptually, choosing hyper-parameters based on predictive metrics may not necessarily lead to good CATE estimation performance, Neal et al. [40] provide empirical evidence that doing so indeed often does in practice.

Table 2 and Table 4 include the hyper-parameter search ranges we set in the SW and TCGA experiments, respectively. Table 3 and Table 5 include the fixed hyper-parameter values across all SW and TCGA experiments, respectively. We restricted the number of hyper-parameter optimization trials to 10 in all experiments. We observed that all models' performances are rather insensitive to hyper-parameter values in the considered search ranges, i.e., the performances across trials have not varied much. The search ranges for the HSIC penalty $\lambda$ are taken from the experimental section of the GraphITE paper [16], where the authors also argue that their model's performance is insensitive to this weight. In consultation with Harada & Kashima [16], we use Ma et al. [34]'s implementation of the normalized HSIC. We use early stopping for all models based on their training loss. We noticed that a patience value below 10 often leads to pre-convergence stopping with subsequent sub-optimal performance for all models but GIN.

### D.2.1 SW

| Hyper-parameter | Search range |
|---|---|
| Num. of layers for covariates representations | 2-4 |
| Num. of layers for treatment representations | 3-6 |
| Num. of layers for $\widehat{m}_{\boldsymbol{\theta}}(\mathbf{X})^{*}$ | 3-6 |
| Num. of layers for $\widehat{e}_{\boldsymbol{\eta}}^{h}(\mathbf{X})^{*}$ | 3-6 |
| Num. of layer for final feed-forward network $^{\dagger}$ | 2-6 |
| Dim. of hidden layers for covariates representations | 50-300 |
| Dim. of hidden layers for treatment representations | 50-300 |
| Dim. of hidden layers for $\widehat{m}_{\boldsymbol{\theta}}(\mathbf{X})^{*}$ | 200-300 |
| Dim. of hidden layers for $\widehat{e}_{\boldsymbol{\eta}}^{h}(\mathbf{X})^{*}$ | 50-150 |
| Dim. of $\widehat{g}_{\boldsymbol{\psi}}(\mathbf{X}), \widehat{h}_{\boldsymbol{\phi}}(\mathbf{T})^{*}$ | 50-250 |
| Dim. of final covariates/treatment layer | 2-200 |
| Dim. of hidden layers for final feed-forward network | 50-300 |
| Num. update steps $K^{*}$ | 10-20 |
| Early stopping patience for $\widehat{m}_{\boldsymbol{\theta}}(\mathbf{X})^{*}$ | {5, 10} |
| Early stopping patience for $\widehat{g}_{\boldsymbol{\psi}}(\mathbf{X}), \widehat{h}_{\boldsymbol{\phi}}(\mathbf{T}), \widehat{e}_{\boldsymbol{\eta}}^{h}(\mathbf{X})^{*}$ | {1, 5} |
| Learning rates $\lambda_{\boldsymbol{\psi}}, \lambda_{\boldsymbol{\phi}}^{*}$ | {5e-4, 1e-3} |
| Learning rate $^{\dagger}$ | {5e-4, 1e-3} |
| Dropout for $\widehat{m}_{\boldsymbol{\theta}}(\mathbf{X})^{*}$ | {0, 0.2} |
| Dropout for $\widehat{e}_{\boldsymbol{\eta}}^{h}(\mathbf{X})^{*}$ | {0, 0.2 } |
| Weight of HSIC penalty $\lambda^{\ddagger}$ | {0.001, 0.01, 1, 10, 100, 1000} |

Table 2: Hyper-parameter search ranges for SW experiments. $^{*}$ denotes hyper-parameter only applicable for GIN; $^{\dagger}$ applicable for all models but GIN, $^{\ddagger}$ applicable only for GraphITE.

| Hyper-parameter | Value |
|---|---|
| Optimizer | Adam [28] |
| Batch size | 500 |
| Weight decay (all optims.) | 0 |
| $\lambda_{\boldsymbol{\theta}}, \lambda_{\boldsymbol{\eta}}$ | 1e-3 |
| Early stopping patience $^{\dagger}$ | 10 |
| GNN Batch Norm | True |
| MLP Batch Norm (all MLPs) | False |
| Activation functions (all layers) | ReLU |
| Validation set size (in %) | 20% |

Table 3: Fixed hyper-parameter values across all SW experiments. $^{*}$ denotes hyper-parameter only applicable for GIN; $^{\dagger}$ applicable for all models but GIN, $^{\ddagger}$ applicable only for GraphITE.

### D.2.2 TCGA

| Hyper-parameter | Search range |
|---|---|
| Num. of layers for covariates representations | 2-5 |
| Num. of layers for treatment representations | 3-6 |
| Num. of layers for $\widehat{m}_{\boldsymbol{\theta}}\left(\mathbf{X}\right)$ * | 2-4 |
| Num. of layers for $\widehat{e}_{\boldsymbol{\eta}}^{h}\left(\mathbf{X}\right)$ * | 1-6 |
| Num. of layer for final feed-forward network $^{\dagger}$ | 1-5 |
| Dim. of hidden layers for covariates representations | 100-400 |
| Dim. of hidden layers for treatment representations | 100-400 |
| Dim. of hidden layers for $\widehat{m}_{\boldsymbol{\theta}}\left(\mathbf{X}\right)$ * | 100-300 |
| Dim. of hidden layers for $\widehat{e}_{\boldsymbol{\eta}}^{h}\left(\mathbf{X}\right)$ * | 10-50 |
| Dim. of $\widehat{g}_{\boldsymbol{\psi}}\left(\mathbf{X}\right),\widehat{h}_{\boldsymbol{\phi}}\left(\mathbf{T}\right)$ * | 200-600 |
| Dim. of final covariates/treatment layer | 2-800 |
| Dim. of hidden layers for final feed-forward network | 100-400 |
| Num. update steps $K$ * | 10-20 |
| Early stopping patience for $\widehat{g}_{\boldsymbol{\psi}}\left(\mathbf{X}\right),\widehat{h}_{\boldsymbol{\phi}}\left(\mathbf{T}\right),\widehat{e}_{\boldsymbol{\eta}}^{h}\left(\mathbf{X}\right)$ * | {5, 10} |
| Learning rates $\lambda_{\psi},\lambda_{\phi}$ * | {5e-4, 1e-3} |
| Learning rate $^{\dagger}$ | {5e-4, 1e-3} |
| Weight of HSIC penalty $\lambda^{\ddagger}$ | {0.001, 0.01, 1, 10, 100, 1000} |

Table 4: Hyper-parameter search ranges for TCGA experiments. * denotes hyper-parameter only applicable for GIN; $^{\dagger}$ applicable for all models but GIN, $^{\ddagger}$ applicable only for GraphITE.

| Hyper-parameter | Value |
|---|---|
| Optimizer | Adam [28] |
| Batch size | 1000 |
| Weight decay (all optims.) | 0 |
| $\lambda_{\boldsymbol{\theta}},\lambda_{\boldsymbol{\eta}}$ | 1e-3 |
| Early stopping patience $^{\dagger}$ | 10 |
| GNN Batch Norm | True |
| MLP Batch Norm (all MLPs) | False |
| Activation functions (all layers) | ReLU |
| Validation set size (in %) | 20% |

Table 5: Fixed hyper-parameter values across all TCGA experiments. * denotes hyper-parameter only applicable for GIN; $^{\dagger}$ applicable for all models but GIN, $^{\ddagger}$ applicable only for GraphITE.

### D.2.3 Hardware details

All experiments were run on Microsoft Azure Virtual Machines with 12 Intel Xeon E5-2690 v4 CPUs and 2 NVIDIA Tesla K80 GPUs. No single trial took longer than $\sim$ 30 minutes to run.

# E Additional Results

## E.1 Comparison of Performances on different $K$ Treatments

We present additional WPEHE@$K$ results for the experiments in Section 6.2 with varying $K$.

| Method | SW | | TCGA | |
|---|---|---|---|---|
| | In-sample | Out-sample | In-sample | Out-sample |
| **WPEHE@2** | | | | |
| Zero | $52.17 \pm 7.37$ | $41.36 \pm 5.04$ | $25.17 \pm 8.12$ | $17.33 \pm 5.41$ |
| CAT | $44.63 \pm 8.18$ | $37.65 \pm 5.90$ | $160.35 \pm 58.56$ | $149.75 \pm 46.86$ |
| GNN | $32.98 \pm 6.63$ | $26.47 \pm 3.87$ | $29.35 \pm 8.90$ | $27.17 \pm 8.67$ |
| GraphITE | $30.18 \pm 6.45$ | $25.39 \pm 4.04$ | $28.60 \pm 9.44$ | $27.37 \pm 9.87$ |
| GIN | $\mathbf{18.00 \pm 3.83}$ | $\mathbf{15.30 \pm 2.60}$ | $\mathbf{10.44 \pm 3.62}$ | $\mathbf{7.76 \pm 1.56}$ |
| **WPEHE@3** | | | | |
| Zero | $51.61 \pm 7.24$ | $41.53 \pm 4.96$ | $25.97 \pm 7.96$ | $17.50 \pm 5.11$ |
| CAT | $44.87 \pm 7.53$ | $37.59 \pm 5.46$ | $159.48 \pm 56.46$ | $148.80 \pm 44.87$ |
| GNN | $32.97 \pm 5.75$ | $26.60 \pm 3.70$ | $30.22 \pm 8.77$ | $27.29 \pm 8.30$ |
| GraphITE | $30.39 \pm 5.89$ | $25.70 \pm 3.70$ | $29.71 \pm 9.43$ | $27.27 \pm 9.38$ |
| GIN | $\mathbf{19.79 \pm 4.06}$ | $\mathbf{15.54 \pm 2.56}$ | $\mathbf{10.62 \pm 3.56}$ | $\mathbf{7.94 \pm 1.51}$ |
| **WPEHE@4** | | | | |
| Zero | $52.92 \pm 7.47$ | $47.93 \pm 6.68$ | $26.35 \pm 7.79$ | $17.76 \pm 5.05$ |
| CAT | $46.95 \pm 7.65$ | $42.47 \pm 6.91$ | $158.02 \pm 54.76$ | $148.08 \pm 43.71$ |
| GNN | $33.89 \pm 5.73$ | $31.51 \pm 5.27$ | $30.51 \pm 8.57$ | $27.53 \pm 8.23$ |
| GraphITE | $31.43 \pm 5.75$ | $30.39 \pm 5.71$ | $30.07 \pm 9.22$ | $27.48 \pm 9.28$ |
| GIN | $\mathbf{20.78 \pm 4.11}$ | $\mathbf{19.50 \pm 4.12}$ | $\mathbf{10.76 \pm 3.51}$ | $\mathbf{8.08 \pm 1.51}$ |
| **WPEHE@5** | | | | |
| Zero | $55.02 \pm 8.00$ | $50.75 \pm 7.92$ | $26.53 \pm 7.66$ | $17.91 \pm 4.96$ |
| CAT | $49.78 \pm 8.37$ | $46.65 \pm 8.86$ | $156.77 \pm 53.58$ | $147.20 \pm 42.86$ |
| GNN | $36.06 \pm 6.69$ | $34.16 \pm 6.41$ | $30.61 \pm 8.41$ | $27.61 \pm 8.10$ |
| GraphITE | $33.69 \pm 6.56$ | $33.13 \pm 6.92$ | $30.22 \pm 9.08$ | $27.53 \pm 9.12$ |
| GIN | $\mathbf{22.06 \pm 4.40}$ | $\mathbf{21.19 \pm 4.80}$ | $\mathbf{10.90 \pm 3.47}$ | $\mathbf{8.13 \pm 1.49}$ |
| **WPEHE@6** | | | | |
| Zero | $56.26 \pm 8.12$ | $53.77 \pm 8.93$ | $26.63 \pm 7.55$ | $17.94 \pm 4.86$ |
| CAT | $51.75 \pm 8.85$ | $49.76 \pm 9.73$ | $155.88 \pm 52.82$ | $146.62 \pm 42.32$ |
| GNN | $37.10 \pm 6.84$ | $36.74 \pm 7.42$ | $30.67 \pm 8.29$ | $27.57 \pm 7.95$ |
| GraphITE | $34.81 \pm 6.70$ | $35.94 \pm 8.07$ | $30.31 \pm 8.96$ | $27.48 \pm 8.95$ |
| GIN | $\mathbf{23.00 \pm 4.56}$ | $\mathbf{23.19 \pm 5.56}$ | $\mathbf{10.98 \pm 3.45}$ | $\mathbf{8.15 \pm 1.46}$ |
| **WPEHE@7** | | | | |
| Zero | $58.16 \pm 8.38$ | $55.73 \pm 9.01$ | $26.66 \pm 7.48$ | $17.97 \pm 4.81$ |
| CAT | $54.62 \pm 9.27$ | $52.21 \pm 9.74$ | $155.24 \pm 52.25$ | $146.15 \pm 41.90$ |
| GNN | $39.21 \pm 7.05$ | $38.51 \pm 7.50$ | $30.67 \pm 8.21$ | $27.56 \pm 7.86$ |
| GraphITE | $37.00 \pm 7.10$ | $37.34 \pm 8.05$ | $30.33 \pm 8.88$ | $27.47 \pm 8.86$ |
| GIN | $\mathbf{24.71 \pm 5.07}$ | $\mathbf{24.46 \pm 5.79}$ | $\mathbf{11.02 \pm 3.43}$ | $\mathbf{8.17 \pm 1.45}$ |
| **WPEHE@8** | | | | |
| Zero | $59.57 \pm 8.74$ | $56.61 \pm 8.94$ | $26.73 \pm 7.43$ | $18.03 \pm 4.76$ |
| CAT | $56.24 \pm 9.71$ | $53.33 \pm 9.71$ | $154.86 \pm 51.85$ | $145.94 \pm 41.61$ |
| GNN | $40.44 \pm 7.36$ | $39.04 \pm 7.33$ | $30.72 \pm 8.16$ | $27.49 \pm 8.78$ |
| GraphITE | $38.42 \pm 7.46$ | $38.06 \pm 7.89$ | $30.39 \pm 8.82$ | $27.49 \pm 8.78$ |
| GIN | $\mathbf{25.90 \pm 5.51}$ | $\mathbf{25.63 \pm 6.03}$ | $\mathbf{11.10 \pm 3.43}$ | $\mathbf{8.20 \pm 1.44}$ |
| **WPEHE@9** | | | | |

| Zero | $60.39 \pm 8.94$ | $55.72 \pm 8.44$ | $26.75 \pm 7.40$ | $18.06 \pm 4.73$ |
|---|---|---|---|---|
| CAT | $57.78 \pm 10.27$ | $53.06 \pm 9.36$ | $154.60 \pm 51.57$ | $145.73 \pm 41.37$ |
| GNN | $41.45 \pm 7.60$ | $38.47 \pm 6.92$ | $30.72 \pm 8.11$ | $27.60 \pm 7.74$ |
| GraphITE | $39.43 \pm 7.69$ | $37.43 \pm 7.48$ | $30.39 \pm 8.78$ | $27.50 \pm 8.72$ |
| GIN | $\mathbf{26.76 \pm 5.80}$ | $\mathbf{25.30 \pm 5.75}$ | $\mathbf{11.12 \pm 3.42}$ | $\mathbf{8.22 \pm 1.43}$ |

| **WPEHE@10** | | | | |
|---|---|---|---|---|
| Zero | $60.92 \pm 9.10$ | $56.44 \pm 8.91$ | $26.78 \pm 7.35$ | $18.09 \pm 4.71$ |
| CAT | $58.32 \pm 10.29$ | $54.76 \pm 10.56$ | $154.39 \pm 51.32$ | $145.57 \pm 41.21$ |
| GNN | $42.08 \pm 7.82$ | $39.11 \pm 7.24$ | $30.73 \pm 8.07$ | $27.61 \pm 7.70$ |
| GraphITE | $40.26 \pm 7.94$ | $37.99 \pm 7.80$ | $30.41 \pm 8.74$ | $27.51 \pm 8.69$ |
| GIN | $\mathbf{27.47 \pm 6.07}$ | $\mathbf{26.01 \pm 6.06}$ | $\mathbf{11.13 \pm 3.41}$ | $\mathbf{8.23 \pm 1.43}$ |

Table 6: Error of CATE estimation for all methods, measured by WPEHE@$1 - 10$. Results are averaged over 10 trials, $\pm$ denotes std. error.

## E.2 Comparison of Robustness to different Bias Strengths $\kappa$

We present additional WPEHE@$K$ results for the experiments in Section 6.3 over increasing bias strength $\kappa$ and varying $K$.

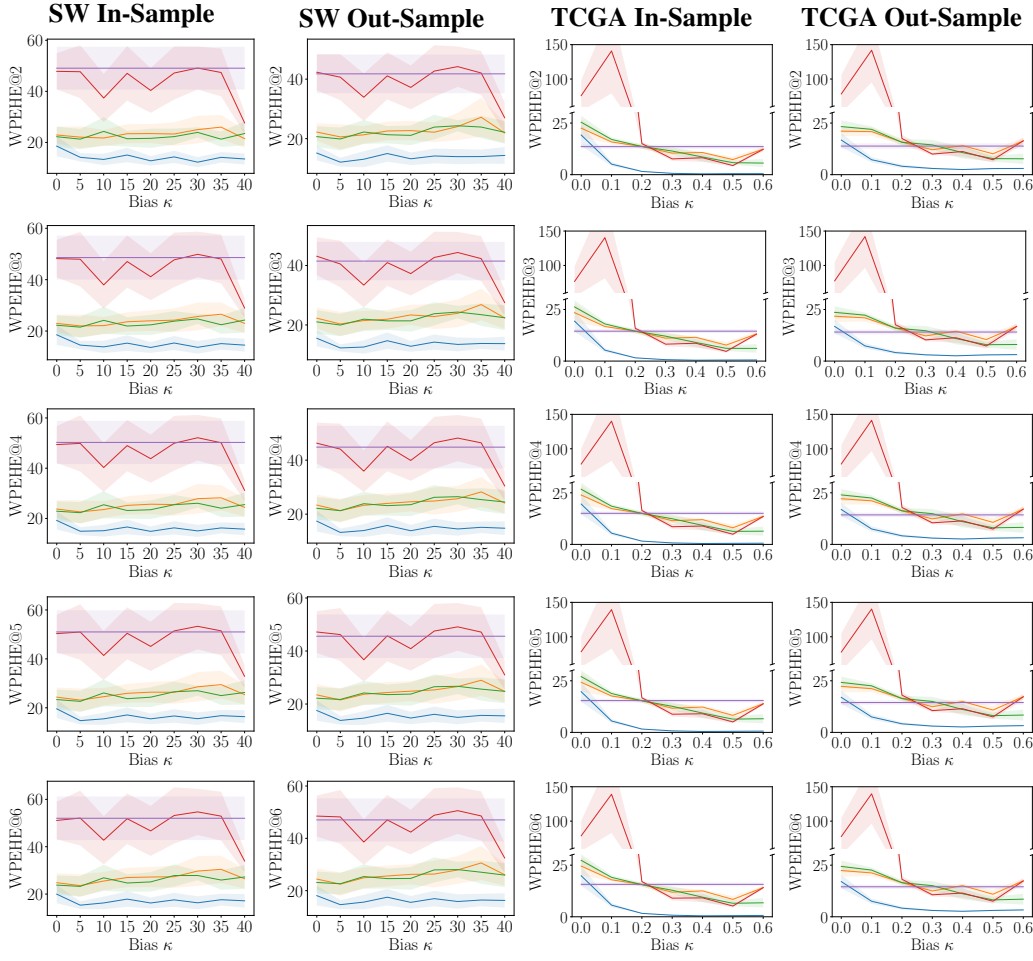

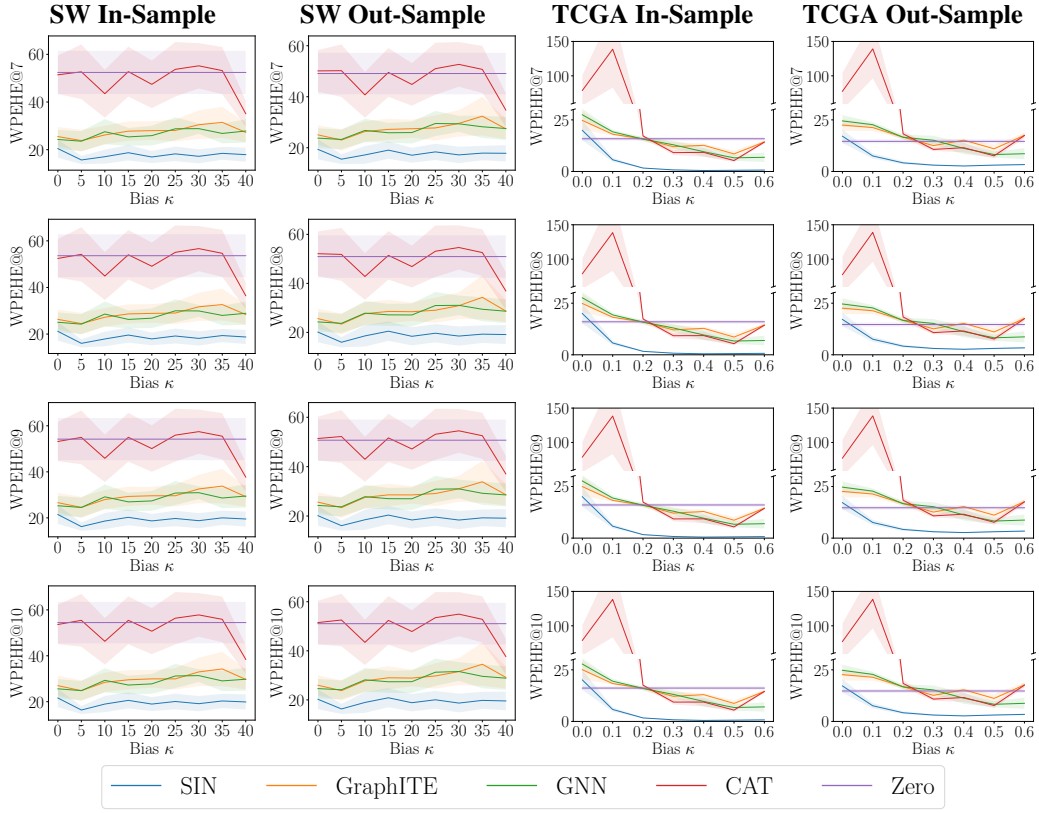

Figure 5: WPEHE@$K$ over increasing bias strength $\kappa$ and varying $K$.

# F    Quasi-oracle rates for generalized R-Learner

The goal of this section is to establish error bounds for learning conditional average treatment effects (CATEs) when treatments are *continuous*. To do so, we will assume that the response function, $\mathbb{E}[Y|\mathbf{X}, \mathbf{T}]$ can be written as follows,

$$\mathbb{E}\left[Y \mid \mathbf{X}, \mathbf{T}\right] = \boldsymbol{\alpha}\left(\mathbf{X}\right)^{\top} \Theta^* \boldsymbol{\beta}\left(\mathbf{T}\right), \tag{33}$$

where $\boldsymbol{\alpha}(\mathbf{X}) \in \mathbb{R}^{d_{\mathbf{X}}}, \boldsymbol{\beta}(\mathbf{T}) \in \mathbb{R}^{d_{\mathbf{T}}}$ are fixed, known basis functions[6] (where $d_{\mathbf{X}}, d_{\mathbf{T}} < \infty$) and $\Theta^* \in \mathbb{R}^{d_{\mathbf{X}} \times d_{\mathbf{T}}}$ is unknown. We will show that we can learn $\Theta^*$ using the generalized Robinson decomposition in eq. (10) (i.e., the minimization is now over $\Theta$) with the same error rate as if we had known the true *oracle* nuisance functions $m^*$ and $e^p$, provided our estimates of $m^*$ and $e^P$ converge to the ground truths at $O(n^{-1/4})$ rate.

The reason we consider the above fixed basis setting instead of the more generic setup in the paper is because there are many things that make the analysis of a more general setup difficult:

- There is a non-trivial dependence between estimators $m(\cdot), e(\cdot), g(\cdot), h(\cdot)$ created by fitting using the entire dataset (as opposed to using cross-fitting).
- Representation learning of the features typically involves non-convex loss functions; the convergence analysis of such is largely still an untackled question.
- In the infinite-basis setting the problem becomes ill-posed (our current work provides insight into fixing this, in particular in Lemma 8).

Addressing these issues is an interesting area of future work. Meanwhile, in this work, we focus on the scenario where the features (i.e. basis functions) are fixed. We first sketch our result without technical jargon as follows.

---

[6]In deep learning jargon, each dimension of the basis functions, $\alpha_i, \beta_j$, is simply called a *feature*.

**Theorem** (Sketch). *Write* $m(\mathbf{x}) := \mathbb{E}\left[Y \mid \mathbf{X} = \mathbf{x}\right]$ *and* $e^P(\mathbf{x}) := \mathbb{E}\left[\boldsymbol{\beta}(\mathbf{T}) \mid \mathbf{X} = \mathbf{x}\right]$. *When the ground truths* $m$ *and* $e^P$ *are unavailable, we can still estimate* $\mathbb{E}\left[Y \mid \mathbf{X}, \mathbf{T}\right]$ *almost with rate* $O\left(n^{-1/2}\right)$ *using only estimates of* $m$ *and* $e^P$, *provided the estimates themselves converge at rate* $O\left(n^{-1/4}\right)$.

## F.1 Preliminaries

To specify the above formally, we follow e.g. [53] to construct an RKHS for the hypothesis space of the response function $f$ as follows. Let $\mathcal{X}$ and $\mathcal{T}$ be compact metric spaces, endowed with finite Borel measures $\mathcal{P}_{\mathcal{X}}$ and $\mathcal{P}_{\mathcal{T}}$. Let $\{\alpha_i\}_{i=1}^{d_\alpha} \subset \mathcal{L}_2(\mathcal{X}, \mathcal{P}_{\mathcal{X}})$ and $\{\beta_i\}_{i=1}^{d_\beta} \subset \mathcal{L}_2(\mathcal{T}, \mathcal{P}_{\mathcal{T}})$ denote subsets of orthonormal functions in $\mathcal{L}_2(\mathcal{X}, \mathcal{P}_{\mathcal{X}})$ and $\mathcal{L}_2(\mathcal{T}, \mathcal{P}_{\mathcal{T}})$ which are feature maps for $\mathbf{X}$ and $\mathbf{T}$, respectively. Write $\boldsymbol{\alpha}, \boldsymbol{\beta} \in \mathbb{R}^{d_\alpha}, \mathbb{R}^{d_\beta}$ as the vectors of features on $\mathbf{X}$ and $\mathbf{T}$, with $\boldsymbol{\alpha}_i(\mathbf{x}) := \alpha_i(\mathbf{x})$ and $\boldsymbol{\beta}_j(\mathbf{t}) := \beta_j(\mathbf{t})$. Then define $k_{\mathbf{X}} : \mathcal{X} \times \mathcal{X} \to \mathbb{R}$ as $k_{\mathbf{X}}(\mathbf{x}_1, \mathbf{x}_2) = \langle \boldsymbol{\alpha}(\mathbf{x}_1), \boldsymbol{\alpha}(\mathbf{x}_2) \rangle_2$ where $\langle \cdot, \cdot \rangle_2$ is the standard Euclidean dot product in $\mathbb{R}^d$, and define similarly $k_{\mathbf{T}} : \mathcal{T} \times \mathcal{T} \to \mathbb{R}$ as $k_{\mathbf{T}}(\mathbf{t}_1, \mathbf{t}_2) = \langle \boldsymbol{\beta}(\mathbf{t}_1), \boldsymbol{\beta}(\mathbf{t}_2) \rangle_2$. Then clearly $k_{\mathbf{X}}$ and $k_{\mathbf{T}}$ are positive definite functions and by Moore-Aronsajn [53, Section 4] there exist unique RKHSes $\mathcal{H}_{\mathcal{X}}, \mathcal{H}_{\mathcal{T}}$ with kernels $k_{\mathbf{X}}$ and $k_{\mathbf{T}}$.

For the readers familiar with [42], we can connect the setup to that of [42] as follows: following e.g. [36], an element $g$ in $\mathcal{H}_{\mathcal{X}}$ can be represented by $g(\mathbf{x}) = \langle \theta, \boldsymbol{\alpha}(\mathbf{x}) \rangle_2 = \langle g, \boldsymbol{\alpha}(\mathbf{x}) \rangle_{\mathcal{H}_{\mathcal{X}}}$. Following [53], we can define an integral operator based on the kernel $k_{\mathbf{X}}$:

$$S_{k_{\mathbf{X}}} : \mathcal{L}_2(\mathcal{X}; \mathcal{P}_{\mathcal{X}}) \to \mathcal{C}(\mathcal{X}) \quad \text{where } \mathcal{C}(\mathcal{X}) \text{ are the continuous functions on } \mathcal{X}. \quad (34)$$

$$(S_{k_{\mathbf{X}}} f)(\mathbf{X}) = \int k_{\mathbf{X}}(\mathbf{x}_1, \mathbf{x}_2) f(\mathbf{x}_2) d\mathcal{P}_{\mathcal{X}}(\mathbf{x}_2), \quad f \in \mathcal{L}_2(\mathcal{X}; \mathcal{P}_{\mathcal{X}}) \quad (35)$$

$$T_{k_{\mathbf{X}}} = I_{k_{\mathbf{X}}} \circ S_{k_{\mathbf{X}}} \quad (36)$$

$$\text{with the inclusion } I_{k_{\mathbf{X}}} : \mathcal{C}(\mathcal{X}) \hookrightarrow \mathcal{L}_2(\mathcal{X}; \mathcal{P}_{\mathcal{X}}) \quad (37)$$

Clearly the eigenfunctions of $T_{k_{\mathbf{X}}}$ are the orthonormal functions $\{\alpha_i\}_{i=1}^{d_\alpha}$ and the non-zero eigenvalues are $\{\sigma_i = 1\}_{i=1}^{d_\alpha}$.

$\mathcal{H}_{\mathcal{T}}$ can be dealt with similarly to $\mathcal{H}_{\mathcal{X}}$. Since $\mathcal{H}_{\mathcal{X} \times \mathcal{T}}$ is isometrically isomorphic to $\mathcal{H}_{\mathcal{X}} \times \mathcal{H}_{\mathcal{T}}$, we can identify the basis functions on $\mathcal{H}_{\mathcal{X} \times \mathcal{T}}$ as $\{\alpha_i \beta_j\}_{i,j=1}^{d_\alpha, d_\beta}$, the eigenvalues as $\{\sigma_{ij} = 1\}_{i,j=1}^{d_\alpha, d_\beta}$, and the inner product $\langle \cdot, \cdot \rangle_{\mathcal{H}_{\mathcal{X} \times \mathcal{T}}} = \langle \cdot, \cdot \rangle_{\mathcal{H}_{\mathcal{X}}} \langle \cdot, \cdot \rangle_{\mathcal{H}_{\mathcal{T}}}$. By construction, the RKHS norm and the $L_2$ norm of $\mathcal{H}_{\mathcal{X} \times \mathcal{T}}$ are both equal to the matrix $2-$norm of the function representer, that is, for $f \in \mathcal{H}_{\mathcal{X} \times \mathcal{T}}, f(\mathbf{x}, \mathbf{t}) = \langle \boldsymbol{\Theta}, \boldsymbol{\alpha}(\mathbf{x}) \otimes \boldsymbol{\beta}(\mathbf{t}) \rangle_2$,

$$\|f\|_{\mathcal{H}_{\mathcal{X} \times \mathcal{T}}} = \|f\|_{L_2} = \|\boldsymbol{\Theta}\|_2 \quad (38)$$

Trivially, for all $0 < p < 1$, the eigenvalues $\sigma_{ij}$ satisfy $G = \sup_{i,j \geq 1}(i + d_{\mathbf{X}}(j-1))^{1/p} \sigma_{ij}$ for some constant $G < \infty$, which was posed as an assumption in [42].

**Remark 1.** *We did not need to require* $\mathcal{X}$ *and* $\mathcal{T}$ *as compact metric spaces. Requiring them to be measurable spaces on which we can define* $\mathcal{L}_2$ *functions should be enough. But compact metric spaces also include most spaces of practical concern, including graph spaces, so we choose it since it satisfies the conditions of Mercer's theorem.*

## F.2 Problem set-up

We assume that the true response function lie in $\mathcal{H}_{\mathcal{X} \times \mathcal{T}}$:

**Assumption 4.** *The true response function* $f^*(\mathbf{x}, \mathbf{t}) = \mathbb{E}[Y \mid \mathbf{X} = \mathbf{x}, \mathbf{T} = \mathbf{t}]$ *can be written as* $f^*(\mathbf{x}, \mathbf{t}) = \boldsymbol{\alpha}^\top(\mathbf{x}) \boldsymbol{\Theta}^* \boldsymbol{\beta}(\mathbf{t})$ *for some matrix of coefficients* $\boldsymbol{\Theta}^*$.

First we write down the population and empirical loss functions we consider. In order to assert that every element of $\mathcal{H}_{\mathcal{X} \times \mathcal{T}}$ can be uniquely represented by some $\boldsymbol{\Theta}$, we use $f_{\boldsymbol{\Theta}}$ to denote $f_{\boldsymbol{\Theta}} := \boldsymbol{\alpha}(\mathbf{x})^T \boldsymbol{\Theta} \boldsymbol{\beta}(\mathbf{t}) \in \mathcal{H}_{\mathcal{X} \times \mathcal{T}}$.

The expected loss of $f_{\boldsymbol{\Theta}}$ is defined by:

$$L(f_{\boldsymbol{\Theta}}) = L(\boldsymbol{\Theta}) = \mathbb{E}\left[\left\{(Y - m^*(\mathbf{X})) - \boldsymbol{\alpha}(\mathbf{X})^T \boldsymbol{\Theta}(\boldsymbol{\beta}(\mathbf{T}) - e^P(\mathbf{X}))\right\}^2\right] \quad (39)$$

The oracle (empirical) loss is defined by:

$$\tilde{L}_n(f_{\boldsymbol{\Theta}}) = \tilde{L}_n(\boldsymbol{\Theta}) = \sum_{l=1}^{n} \left[ \left\{ (Y - m^*(\mathbf{X}_l)) - \boldsymbol{\alpha}(\mathbf{X}_l)^T \boldsymbol{\Theta}(\boldsymbol{\beta}(\mathbf{T}_l) - e^P(\mathbf{X}_l)) \right\}^2 \right] \tag{40}$$

The feasible (empirical) loss is defined by:

$$\hat{L}_n(f_{\boldsymbol{\Theta}}) = \hat{L}_n(\boldsymbol{\Theta}) = \sum_{l=1}^{n} \left[ \left\{ (Y - \hat{m}(\mathbf{X}_l)) - \boldsymbol{\alpha}(\mathbf{X}_l)^T \boldsymbol{\Theta}(\boldsymbol{\beta}(\mathbf{T}_l) - \hat{e}^P(\mathbf{X}_l)) \right\}^2 \right] \tag{41}$$

Note that we use $L(f_{\boldsymbol{\Theta}})$ and $L(\boldsymbol{\Theta})$ interchangeably due to the bijection between $\boldsymbol{\Theta} \in \mathbb{R}^{d_{\mathbf{X}} \times d_{\mathbf{T}}}$ and $\mathcal{H}_{\mathcal{X} \times \mathcal{T}}$.

The corresponding regret functions are defined by

$$R(\boldsymbol{\Theta}) = L(\boldsymbol{\Theta}) - L(\boldsymbol{\Theta}^*) \tag{42}$$

$$\tilde{R}_n(\boldsymbol{\Theta}) = \tilde{L}_n(\boldsymbol{\Theta}) - \tilde{L}_n(\boldsymbol{\Theta}^*) \tag{43}$$

$$\hat{R}_n(\boldsymbol{\Theta}) = \hat{L}_n(\boldsymbol{\Theta}) - \hat{L}_n(\boldsymbol{\Theta}^*) \tag{44}$$

We now formally state the assumptions we need to derive the result in Theorem 2.

**Assumption 5** (Overlap). *The marginal distribution of features $\mathcal{P}_{\boldsymbol{\alpha}(\mathcal{X})\boldsymbol{\beta}(\mathcal{T})}$ is positive, i.e.* $\mathrm{supp}[\mathcal{P}_{\boldsymbol{\alpha}(\mathcal{X})\boldsymbol{\beta}(\mathcal{T})}] = \boldsymbol{\alpha}(\mathcal{X})\boldsymbol{\beta}(\mathcal{T})$.

**Assumption 6** (Boundedness). *Without loss of generality, we assume that for all $\mathbf{X} \in \mathcal{X}, \mathbf{T} \in \mathcal{T}$, $\sup_i \|\alpha_i(\mathbf{X})\|_\infty, \sup_j \|\beta_j(\mathbf{T})\|_\infty \leq A < \infty$. We also assume that the outcome $Y$ are almost surely bounded, i.e. $\mathbb{P}(|Y| < B < \infty) = 1$.*

For clarity, we list all notations we use here.

**Notation.**

- $\mathcal{H}$: A Product Reproducing Kernel Hilbert Space with finite number of basis functions, with $\boldsymbol{\alpha}$ the features of $X$ and $\boldsymbol{\beta}$ the features of $T$.
- $\Theta$: The matrix of coefficients for a given function in $\mathcal{H}$.
- $f_{\Theta}$: $f_{\Theta}(X,T) := \boldsymbol{\alpha}(X)^\top \Theta \boldsymbol{\beta}(T)$.
- $\mathcal{H}_c$: The subset of $\mathcal{H}$ which is the ball of radius $c$.
- $\Theta_c$: $f_{\Theta_c}$ is a minimiser of the loss in $\mathcal{H}_c$.
- $R(f_{\Theta})$: $L(f_{\Theta}) - L(f^*)$.
- $R(f_{\Theta}; c)$: $L(f_{\Theta}) - L(f_{\Theta_c})$

**Convention.** Throughout, we will use capital letters $A, B, C, ...,$ possibly with subscripts and superscripts, e.g. $A_1, B^{(2)}$, etc. to denote constants. We may overload notation and use the same letter to denote different constants.

### F.3 Proof strategy

Here we lay forward the detailed proof for the quasi-oracle convergence rate for a featurized continuous heterogeneous treatment effect estimation algorithm with Robinson decomposition. Our proof extends the structure of Nie & Wager [42]. To make the proof self-contained while simultaneously highlighting the differences with Nie & Wager [42], we present a complete version of the proof, where we will pause to describe any difference and its significance where it appears.

The high-level idea of showing 'quasi-oracle' error rate is as follows. First, we show that both the feasible loss and the oracle loss satisfy the same (quasi-)isomorphism with the true loss, where the tightness of the quasi-isomorphism increases as sample size increases. The quasi-isomorphism with the true loss then leads us to bound the feasible and oracle losses by the same quantity, which decreases to 0 as sample size grows indefinitely. To show the (quasi-)isomorphism for the oracle

learner can be done by leveraging on the standard least-squares regression ideas [36]; to achieve the same for the feasible learner relies on the fact that the feasible loss differs from the oracle loss by only a small amount relative to the true loss, which constitutes the bulk of the proof.

We start with stating the formal lemma which connects quasi-isomorphism with loss bounds.

### F.4 From quasi-isomorphism to regret bound

**Definition 3** (loss function). *A function is a **loss function** if it maps from a hypothesis class $\mathcal{H}$, to the real numbers $\mathbb{R}$.*

**Lemma 4.** *Let $\check{L}(f_\Theta \in \mathcal{H}_c)$ be a loss function, and $\check{R}(f_\Theta; c) = \check{L}(f_\Theta) - \check{L}(f_{\Theta_c})$ be the associated c-regret. Suppose $\rho(r)$ is a positive, continuous, increasing function. If, $\forall\, 1 \le c \le C$ and some $k > 1$, the following inequality holds for all $f_\Theta \in \mathcal{H}_c$:*

$$\frac{1}{k}\check{R}(f_\Theta; c) - \rho(c) \le R(f_\Theta; c) \le k\check{R}(f_\Theta; c) + \rho(c) \tag{45}$$

*Then, writing $\kappa_1 = 2k + \frac{1}{k}$ and $\kappa_2 = 2k^2 + 3$, any solution to the regularized minimization problem with $\Lambda(c) \ge \rho(c)$,*

$$f_{\tilde{\Theta}} \in \arg\min_{f_\Theta \in \mathcal{H}_C}\{\check{L}(f_\Theta) + \kappa_1\Lambda(f_\Theta)_\mathcal{H}\} \tag{46}$$

*also satisfied the following risk bound:*

$$L(f_{\tilde{\Theta}}) \le \inf_{f_\Theta \in \mathcal{H}_C}\{L(f_\Theta) + \kappa_2\Lambda(f_\Theta)_\mathcal{H} \tag{47}$$

*Proof.* Notice that $\{\mathcal{H}_c; c \ge 1\}$ is an ordered set. Thus the same argument as [42] applies. $\qquad\square$

Lemma 4 tells us that if we have a quasi-isomorphism of the regrets in the form of 45, we immediately can bound the expected risk of the (regularized) minimizer of the corresponding loss, $\check{L}$ as in 47.

### F.5 A concrete instance of $\rho(c)$ satisfying 45

By setting $\check{R}$ to $\tilde{R}_n$, 4 gives us a way to bound the oracle regret, but we still need a concrete formulation of $\rho(c)$ to derive the oracle convergence rate. To this end, we may use the result of Mendelson & Neeman [36], but first we must show that their results can be applied to our setting.

Mendelson & Neeman [36] consider the optimization over a space of RKHS functions with the least-squares loss. Our oracle case can be thought of in the same way as follows: since $m^*$ is an oracle quantity, $Y - m^*(\mathbf{X})$ can be thought of as the labels, the space $\overline{\mathcal{H}} = \{f_\Theta : \mathcal{X} \times \mathcal{T} \to R;$ for some $\Theta \in \mathbb{R}^{d_\mathbf{X} \times d_\mathbf{T}}, f_\Theta(\mathbf{x}, \mathbf{t}) = \boldsymbol{\alpha}(\mathbf{x})^\top\Theta(\boldsymbol{\beta}(\mathbf{t}) - e^P(\mathbf{x}))\}$ is an RKHS with features $\boldsymbol{\alpha}(\mathbf{X}) \otimes (\boldsymbol{\beta}(\mathbf{T}) - e^P(\mathbf{X}))$. Thus, our setting can be thought of as a least-squares optimization over the RKHS $\overline{\mathcal{H}}$ and the results from [36] applies. To use the results of [36], we still need the following technical result which decomposes $\overline{\mathcal{H}}$ into an *ordered, parameterized hierarchy*.

**Definition 5** (Ordered, parameterized hierarchy). *As defined in [36], let $\mathcal{F}$ be a class of functions and suppose that there is a collection of subsets $\{\mathcal{F}_r; r \ge 1\}$ with the following properties:*

1. *$\{\mathcal{F}_r : r \ge 1\}$ is monotone (i.e. whenever $r \le s, \mathcal{F}_r \subseteq \mathcal{F}_s$);*

2. *for every $r \ge 1$, there exists a unique element $f_r^* \in \mathcal{F}_r$ such that $L(f_r^*) = \inf_{f \in \mathcal{F}_r} L(f)$;*

3. *the map $r \to L(f_r^*)$ is continuous;*

4. *for every $r_0 \ge 1, \bigcap_{r \le r_0} \mathcal{F}_r = \mathcal{F}_{r_0}$;*

5. *$\bigcup_{r \le 1} \mathcal{F}_r = \mathcal{F}$.*

*Given a class of functions $\mathcal{F}$, we say that $\{\mathcal{F}_r; r \ge 1\}$ is an **ordered, parameterized hierarchy of $\mathcal{F}$** if the above conditions 1-5 are satisfied.*

**Lemma 6.** *Define*

$$\overline{\mathcal{H}_c} := \{f_\Theta : \mathcal{X} \times \mathcal{T} \to \mathbb{R} : \exists \Theta, \|\Theta\|_2 \le c, \tag{48}$$

$$s.t. \quad f_\Theta(\mathbf{X}, \mathbf{T}) = \boldsymbol{\alpha}(\mathbf{X})^\top \Theta(\boldsymbol{\beta}(\mathbf{T}) - e^p(\mathbf{X}))\}, \tag{49}$$

*then* $\left\{ \overline{\mathcal{H}_c} \right\}_{1 \le c \le C}$ *is an ordered parameterized hierarchy.*

*Proof.* The first, fourth and fifth properties follow immediately. $\overline{\mathcal{H}_c}$ is clearly convex. It is compact because every sequence $\{f_{\Theta_i}\}_i \subset \overline{\mathcal{H}_c}$ is induced by $\{\Theta_i\}_i \subset \mathbb{R}^n, \|\Theta_i\|_2 \le c$, and by Bolzano-Weierstrass theorem in $\mathbb{R}^n$, every bounded sequence has a convergent subsequence $\{\Theta_k\}_k \subset \{\Theta_i\}_i$ (w.r.t. the Euclidean norm). Thus pick the $N$ such that for all $k \ge N$ where $\|\Theta_k - \Theta_N\|_2 \le \epsilon$, and then

$$\left\| f_{\Theta_k} - f_{\Theta_N} \right\|_{L_2(P(\mathcal{X}, \mathcal{T}))} = \left\| f_{\Theta_k - \Theta_N} \right\|_{L_2(P(\mathcal{X}, \mathcal{T}))} = \mathbb{E} \left[ \langle \Theta_k - \Theta_N, \boldsymbol{\alpha}(\mathbf{X}) \otimes (\boldsymbol{\beta}(\mathbf{T}) - e^p(\mathbf{X})) \rangle^2 \right]^{1/2} \tag{50}$$

$$\le \mathbb{E} \left[ \|\Theta_k - \Theta_N\|_2 \|\boldsymbol{\alpha}(\mathbf{X}) \otimes (\boldsymbol{\beta}(\mathbf{T}) - e^p(\mathbf{X}))\|_2 \right]^{1/2} \tag{51}$$

$$\le \epsilon \mathbb{E} \left[ \|\boldsymbol{\alpha}(\mathbf{X}) \otimes (\boldsymbol{\beta}(\mathbf{T}) - e^p(\mathbf{X}))\|_2 \right]^{1/2} \le \epsilon B, \tag{52}$$

where $\left\| \boldsymbol{\alpha}(\mathbf{X}) \otimes (\boldsymbol{\beta}(\mathbf{T}) - e^p(\mathbf{X})) \right\|_2 \le B$ by Assumption 6 for some constant B. The second property now follows from the fact that $\overline{\mathcal{H}_c}$ is convex and compact. The third property follows by the same argument as [36]. $\square$

Mendelson & Neeman [36] thus provides a formulation of $\rho$ which, with some constant $U(\epsilon)$, for large enough $n$ and probability at least $1 - \epsilon$, satisfies 120 for the oracle loss function $\tilde{R}_n$ with $k = 2$:

$$\rho_n(c) = U(\epsilon) \left\{ 1 + \log(n) + \log\left(\log(c+e)\right) \right\} \left( \frac{(c+1)^p \log(n)}{\sqrt{n}} \right)^{2/(1+p)} \tag{53}$$

Thus, we may now realize the convergence rate for the oracle learner as follows.

### F.6    Oracle convergence rate.

With 53, Lemma 4 immediately implies that penalized regression over $\mathcal{H}_C$ with the oracle loss function $\tilde{L}_n(\cdot)$ and regularizer $\kappa_1 \rho_n(c)$ satisfies the bound below with high probability:

$$R(\tilde{\Theta}_n) = L(\tilde{\Theta}_n) - L(\Theta^*) \le \inf_{\Theta \in \mathcal{H}_C} \left\{ L((\Theta) + \kappa_2 \rho_n(\|\Theta\|_{\mathcal{H}}) \right\} - L(\Theta^*) \tag{54}$$

Furthermore, Corollary 2.7 in [36] gives that for any $1 < c < C$,

$$\inf_{\Theta \in \mathcal{H}_C} \left\{ L(\Theta) + \kappa_2 \rho_n(\|\Theta\|_{\mathcal{H}}) \right\} \le L(\Theta^*) + \left\{ L(\Theta_c^*) - L(\Theta^*) \right\} + \kappa_2 \rho_n(c) \tag{55}$$

Finally, note that for large enough $c$,

$$\left\{ L\left(\Theta_c^*\right) - L\left(\Theta^*\right) \right\} = 0, \tag{56}$$

so the error is dominated by $\rho_n(c)$, at

$$R\left(\tilde{\Theta}_n\right) = \mathcal{O}\left( (\log(n))^{\frac{3+p}{1+p}} n^{-\frac{1}{1+p}} \right) = \tilde{\mathcal{O}}(n^{-\frac{1}{1+p}}), \tag{57}$$

where $\tilde{\mathcal{O}}$ notation ignores the logarithmic factors.

### F.7 Bridging $\hat{R}_n$ and $\tilde{R}_n$

Now that we have the oracle convergence rate, we show a bridging result which will let us conclude that 45 holds for $\hat{R}_n$ as well, and thus the oracle rate also holds for $\hat{R}_n$.

To yield that bridging result, we first need to leverage the assumption of overlap to relate the $L_2$ difference between $f_\Theta$ and $f_{\Theta_c}$, i.e. $\mathbb{E}\left[\left(f_\Theta(\mathbf{X}, \mathbf{T}) - f_{\Theta_c}(\mathbf{X}, \mathbf{T})\right)^2\right]$, with the $c-$regret $R(\Theta; c)$. We first show that the $L_2$ difference is always upper bounded by the regret up to a constant.

**Lemma 7.** $\exists \epsilon > 0$ s.t. for all $f \in \mathcal{H}_c$, $\mathbb{E}_\alpha[\langle f, \alpha \rangle^2] \geq \epsilon \|f\|_{L_2}$ where $\alpha$ is a r.v. taking values in $\mathcal{H}_c$ and the support of $\alpha$ is of Lebesgue-measure non-zero in $\mathcal{H}_c$.

*Proof.* Let $S = \{f \in \mathcal{H}_c : \|f\|_{\mathcal{H}_c} = 1\}$, and define $g : S \to \mathbb{R}^+$ as $g(f) = \mathbb{E}_\alpha[\langle f, \alpha \rangle^2]$. By Jensen's inequality, $\mathbb{E}_\alpha[\langle f, \alpha \rangle^2] \geq 0$ since $\langle f, \cdot \rangle^2 : \alpha \mapsto \langle f, \alpha \rangle^2$ is a convex function in $\alpha$. Moreover, whenever $supp[\mathcal{P}_\alpha]$ is Lebesgue-measure non-zero in $\mathcal{H}_c$, $\langle f, \cdot \rangle^2$ is non-linear on $supp[\mathcal{P}_\alpha]$, so the inequality is strict:

$$\mathbb{E}_\alpha[\langle f, \alpha \rangle^2] > 0. \tag{58}$$

Now since $\mathcal{H}_c$ is finite-dimensional, $S$ is compact. Since $g$ is continuous in $f$, and the continuous image of a compact set is compact, we have that $g(S)$ is compact, and therefore closed.

Note, at this point, that $g(S)$ is the set of values achieved by $\mathbb{E}_\alpha[\langle f, \alpha \rangle^2]$ at various values of $f$. By equation 58, $g(S) \not\ni 0$. Since $g(S)$ is compact, its complement thus contains 0. Moreover, since $\mathbb{R}^+ \setminus g(S) \ni 0$, $\exists$ a ball around 0 of radius $\tilde{\epsilon} > 0$ s.t. $[0, \tilde{\epsilon}) \subset \mathbb{R}^+ \setminus g(S)$. Therefore, $g(S) \subset \mathbb{R}^+$ is lower bounded by $\tilde{\epsilon} > 0$.

Therefore,

$$\forall f \in \mathcal{H}_c, \mathbb{E}_\alpha\left[\langle f, \alpha \rangle^2\right] = \|f\|_{\mathcal{H}_c}^2 \mathbb{E}_\alpha\left[\left\langle \frac{f}{\|f\|_{\mathcal{H}_c}^2}, \alpha \right\rangle^2\right] \geq \epsilon \|f\|_{\mathcal{H}_c}^2 = \epsilon \|f\|_{L_2}^2, \tag{59}$$

for some $\epsilon > 0$. The last inequality is due to 38. $\qquad \square$

**Lemma 8** (Usage of the overlap condition in the multiple treatment setting). *Under Assumption 5, i.e. we have overlap on the features, that is $supp[\mathcal{P}_{\alpha(\mathcal{X}) \times \beta(\mathcal{T})}] = \alpha(\mathcal{X}) \times \beta(\mathcal{T})$, then $\exists A \in \mathbb{R}$ s.t.*

$$\mathbb{E}[\left(f_\Theta(X, T) - f_{\Theta_c}(X, T)\right)^2] < AR(\Theta; c) \tag{60}$$

*Proof.* Within $\mathcal{H}_c$, we seek to upper bound excess $L_2$ risk of $f_\Theta$ by its c-regret $R(\Theta; c)$; $R(\Theta; c) = L(\Theta) - L(\Theta_c)$.

First we write down the expected loss functional again:

$$L(\boldsymbol{\Theta}) = \mathbb{E}[(\{Y - m^*(\mathbf{X})\} - \{f_\Theta(\mathbf{X}.\mathbf{T}) - \mathbb{E}[f_\Theta(\mathbf{X}, \mathbf{T})|\mathbf{X}]\})^2] \tag{61}$$
$$= \mathbb{E}[\mathbb{V}\{Y - m^*(\mathbf{X})|\mathbf{X}, \mathbf{T}\}] + \mathbb{E}[\{(f^*(\mathbf{X}, \mathbf{T}) - f_\Theta(\mathbf{X}, \mathbf{T})) - \mathbb{E}[f^*(\mathbf{X}, \mathbf{T}) - f_\Theta(\mathbf{X}, \mathbf{T}) \mid \mathbf{X}]\}^2] \tag{62}$$

Thus the regret of $\boldsymbol{\Theta}$, which is defined as $L(\boldsymbol{\Theta}) - L(f^*)$, is:

$$R(\boldsymbol{\Theta}) = \mathbb{E}\left[\{(f^*(\mathbf{X}, \mathbf{T}) - f_\Theta(\mathbf{X}, \mathbf{T}) - \mathbb{E}[f^*(\mathbf{X}, \mathbf{T}) - f_\Theta(\mathbf{X}, \mathbf{T}) \mid \mathbf{X}]\}^2\right] \tag{63}$$
$$= \mathbb{E}[\{(f_\Theta(\mathbf{X}, \mathbf{T}) - f_{\Theta_c}(\mathbf{X}, \mathbf{T})) - \mathbb{E}[f_\Theta(\mathbf{X}, \mathbf{T}) - f_{\Theta_c}(\mathbf{X}, \mathbf{T}) \mid \mathbf{X}]$$
$$+ (f_{\Theta_c}(\mathbf{X}, \mathbf{T}) - f^*(\mathbf{X}, \mathbf{T})) - \mathbb{E}[f_{\Theta_c}(\mathbf{X}, \mathbf{T}) - f^*(\mathbf{X}, \mathbf{T}) \mid \mathbf{X}]\}^2] \tag{64}$$
$$= \mathbb{E}[\{f_\Theta(\mathbf{X}, \mathbf{T}) - f_{\Theta_c}(\mathbf{X}, \mathbf{T})) - \mathbb{E}[f_\Theta(\mathbf{X}, \mathbf{T}) - f_{\Theta_c}(\mathbf{X}, \mathbf{T}) \mid \mathbf{X}]\}^2]$$
$$+ \mathbb{E}[\{f_{\Theta_c}(\mathbf{X}, \mathbf{T}) - f^*(\mathbf{X}, \mathbf{T}) - \mathbb{E}[f_{\Theta_c}(\mathbf{X}, \mathbf{T}) - f^*(\mathbf{X}, \mathbf{T}) \mid \mathbf{X}]\}^2]$$
$$+ 2\mathbb{E}[\{(f_\Theta(\mathbf{X}, \mathbf{T}) - f_{\Theta_c}(\mathbf{X}, \mathbf{T})) - \mathbb{E}[f_\Theta(\mathbf{X}, \mathbf{T}) - f_{\Theta_c}(\mathbf{X}, \mathbf{T}) \mid \mathbf{X}]\}$$
$$\cdot \{(f_{\Theta_c}(\mathbf{X}, \mathbf{T}) - f^*(\mathbf{X}, \mathbf{T})) - \mathbb{E}[f_{\Theta_c}(\mathbf{X}, \mathbf{T}) - f^*(\mathbf{X}, \mathbf{T}) \mid \mathbf{X}]\}] \tag{65}$$

Note that, by definition the c-regret of $\boldsymbol{\Theta}$ is just the difference between the regret of $\boldsymbol{\Theta}$ and $\boldsymbol{\Theta}_c$. And the regret of $\boldsymbol{\Theta}_c$ is the second term in equation 65. Thus, the c-regret of $\boldsymbol{\Theta}$ is the first and third term of equation 65.

Now, note that the third term is non-negative because $\mathcal{H}_c$ is convex. To see this, note that it is equal to

$$\frac{\partial}{\partial \epsilon} R\left(\boldsymbol{\Theta}_c + \epsilon\left(\boldsymbol{\Theta} - \boldsymbol{\Theta}_c\right)\right)\mid_{\epsilon=0}, \tag{66}$$

which must be non-negative for any $\boldsymbol{\Theta} \in \mathcal{H}_c$ since otherwise there will be another point in $\mathcal{H}_c$ which has a smaller regret than $\boldsymbol{\Theta}_c$.

Therefore,

$$R(\boldsymbol{\Theta}; c) \geq \mathbb{E}\left[\left\{f_{\boldsymbol{\Theta}}(\mathbf{X}, \mathbf{T}) - f_{\boldsymbol{\Theta}_c}(\mathbf{X}, \mathbf{T})) - \mathbb{E}\left[f_{\boldsymbol{\Theta}}(\mathbf{X}, \mathbf{T}) - f_{\boldsymbol{\Theta}_c}(\mathbf{X}, \mathbf{T}) \mid \mathbf{X}\right]\right\}^2\right] \tag{67}$$

$$= \mathbb{E}\left[\left\{\boldsymbol{\alpha}(\mathbf{X})^\top(\boldsymbol{\Theta} - \boldsymbol{\Theta}_c)(\boldsymbol{\beta}(\mathbf{T}) - e^p(\mathbf{X}))\right\}^2\right] \tag{68}$$

$$= \mathbb{E}\left[\left\langle \boldsymbol{\Theta} - \boldsymbol{\Theta}_c, \boldsymbol{\alpha}(\mathbf{X}) \otimes (\boldsymbol{\beta}(\mathbf{T}) - e^p(\mathbf{X}))\right\rangle^2\right] \tag{69}$$

Now, we would like to show that $\mathbb{E}\left[\left\langle \boldsymbol{\Theta} - \boldsymbol{\Theta}_c, \boldsymbol{\alpha}(\mathbf{X}) \otimes (\boldsymbol{\beta}(\mathbf{T}) - e^p(\mathbf{X}))\right\rangle^2\right]$ is bounded below by the norm of $\boldsymbol{\Theta} - \boldsymbol{\Theta}_c$ up to some multiplicative constant. We do so using Lemma 7. Under the context of Lemma 7, set $f := \boldsymbol{\Theta} - \boldsymbol{\Theta}_c$, and $\alpha := \boldsymbol{\alpha}(\mathbf{X}) \otimes (\boldsymbol{\beta}(\mathbf{T}) - e^p(\mathbf{X}))$. To check that the support of $\alpha$ is not of measure 0, we first note that the support of $\boldsymbol{\alpha}(\mathbf{X})$ is not measure 0 by assumption; secondly, the support of $\boldsymbol{\beta}(\mathbf{T}) - e^p(\mathbf{X})$ is not measure 0 provided that $P(\boldsymbol{\beta}(T) \mid X)$ is a positive measure for any $X$. Then by Lemma 7, we have that $\exists \epsilon > 0$

$$R(\boldsymbol{\Theta}; c) \geq \epsilon \|f_{\boldsymbol{\Theta}} - f_{\boldsymbol{\Theta}_c}\|_{L_2} \tag{70}$$

$\square$

Immediately after Lemma 8, we derive a bound on the infinity norm using the regret function which we will repeatedly use later.

**Corollary 9.** *Following from 38 and Lemma 8,*

$$\|\boldsymbol{\Theta} - \boldsymbol{\Theta}_c\|_\infty \leq const(p)\|f_{\boldsymbol{\Theta}} - f_{\boldsymbol{\Theta}_c}\|_{\mathcal{H}}^p \|f_{\boldsymbol{\Theta}} - f_{\boldsymbol{\Theta}_c}\|_{L_2}^{1-p} \leq const(p)c^p R(\boldsymbol{\Theta}; c)^{\frac{1-p}{2}} \tag{71}$$

*where we note that the second inequality follows from combining Lemma 8 with the fact that for* $f_{\boldsymbol{\Theta}} \in \mathcal{H}_c, \|f_{\boldsymbol{\Theta}} - f_{\boldsymbol{\Theta}_c}\| \leq 2c$ *by the triangle inequality.*

*Proof.* Immediate from 38 and Lemma 8. $\square$

Using Lemma 8, we can further show that the $L_2$ difference between two constrained optima only depends on the $L_2$ norm of the one with the weaker constraint.

**Corollary 10.** *Suppose we have overlap, i.e. Assumption 5. Then with a positive constant const. $> 0$, the following holds for $1 < c < c'$.*

$$\|f_{\boldsymbol{\Theta}_c} - f_{\boldsymbol{\Theta}_{c'}}\|_{L_2} \leq const.\|f_{\boldsymbol{\Theta}_{c'}}\|_{L_2} \tag{72}$$

*Proof.* We have shown that

$$R(\boldsymbol{\Theta}; c) \geq \epsilon \|f_{\boldsymbol{\Theta}} - f_{\boldsymbol{\Theta}_c}\|_{L_2}^2 \tag{73}$$

Then following [42], we check that

$$\left\|\boldsymbol{\Theta}_c - \frac{c}{c'}\boldsymbol{\Theta}_{c'}\right\|_{L_2}^2 \leq \epsilon R(\frac{c}{c'}\boldsymbol{\Theta}_{c'}; c) \tag{74}$$

$$= \epsilon\left(L(\frac{c}{c'}\boldsymbol{\Theta}_{c'}) - L(\boldsymbol{\Theta}_c)\right) \tag{75}$$

$$\leq \epsilon\left(L(\frac{c}{c'}\boldsymbol{\Theta}_{c'}) - L(\boldsymbol{\Theta}_{c'})\right) \tag{76}$$

$$= \epsilon\left(R(\frac{c}{c'}\boldsymbol{\Theta}_{c'}) - R(\boldsymbol{\Theta}_{c'})\right) \tag{77}$$

To bound $R\left(\frac{c}{c'}\mathbf{\Theta}_{c'}\right) - R(\mathbf{\Theta}_{c'})$, note

$$
R(\mathbf{\Theta}) = \mathbb{E}\left[\{(f_{\mathbf{\Theta}}(\mathbf{X},\mathbf{T}) - f_{\mathbf{\Theta}_{c'}}(\mathbf{X},\mathbf{T})) - \mathbb{E}[f_{\mathbf{\Theta}}(\mathbf{X},\mathbf{T}) - f_{\mathbf{\Theta}_{c'}}(\mathbf{X},\mathbf{T}) \mid \mathbf{X}]\}^2\right]
$$
$$
+ \mathbb{E}\left[\{f_{\mathbf{\Theta}_{c'}}(\mathbf{X},\mathbf{T}) - f^*(\mathbf{X},\mathbf{T}) - \mathbb{E}[f_{\mathbf{\Theta}_{c'}}(\mathbf{X},\mathbf{T}) - f^*(\mathbf{X},\mathbf{T}) \mid \mathbf{X}]\}^2\right]
$$
$$
+ 2\mathbb{E}\left[\left\{(f_{\mathbf{\Theta}}(\mathbf{X},\mathbf{T}) - f_{\mathbf{\Theta}_{c'}}(\mathbf{X},\mathbf{T})) - \mathbb{E}\left[f_{\mathbf{\Theta}}(\mathbf{X},\mathbf{T}) - f_{\mathbf{\Theta}_{c'}}(\mathbf{X},\mathbf{T}) \mid \mathbf{X}\right]\right\}\right.
$$
$$
\left. \cdot \left\{(f_{\mathbf{\Theta}_{c'}}(\mathbf{X},\mathbf{T}) - f^*(\mathbf{X},\mathbf{T})) - \mathbb{E}\left[f_{\mathbf{\Theta}_{c'}}(\mathbf{X},\mathbf{T}) - f^*(\mathbf{X},\mathbf{T}) \mid \mathbf{X}\right]\right\}\right] \tag{78}
$$

so $R(\mathbf{\Theta}_{c'})$ is just the second term of equation 78, which we drop when considering $R(\frac{c}{c'}\mathbf{\Theta}_{c'}) - R(\mathbf{\Theta}_{c'})$

$$
R\left(\frac{c}{c'}\mathbf{\Theta}_{c'}\right) - R(\mathbf{\Theta}_{c'}) = \mathbb{E}\left[\{(\frac{c}{c'} - 1)f_{\mathbf{\Theta}_{c'}}(\mathbf{X},\mathbf{T}) - \mathbb{E}[(\frac{c}{c'} - 1)f_{\mathbf{\Theta}_{c'}}(\mathbf{X},\mathbf{T}) \mid \mathbf{X}]\}^2\right]
$$
$$
+ 2\mathbb{E}\left[\left\{(\frac{c}{c'} - 1)f_{\mathbf{\Theta}_{c'}}(\mathbf{X},\mathbf{T}) - \mathbb{E}\left[(\frac{c}{c'} - 1)f_{\mathbf{\Theta}_{c'}}(\mathbf{X},\mathbf{T}) \mid \mathbf{X}\right]\right\}\right.
$$
$$
\left. \cdot \left\{(f_{\mathbf{\Theta}_{c'}}(\mathbf{X},\mathbf{T}) - f^*(\mathbf{X},\mathbf{T})) - \mathbb{E}\left[f_{\mathbf{\Theta}_{c'}}(\mathbf{X},\mathbf{T}) - f^*(\mathbf{X},\mathbf{T}) \mid \mathbf{X}\right]\right\}\right] \tag{79}
$$
$$
= \mathbb{E}\left[\left\{\boldsymbol{\alpha}(\mathbf{X})^\top \left(\frac{c}{c'} - 1\right)\mathbf{\Theta}_{c'}\left(\boldsymbol{\beta}(\mathbf{T}) - e^p(\mathbf{X})\right)\right\}^2\right]
$$
$$
+ 2\mathbb{E}\left[\left\{\boldsymbol{\alpha}(\mathbf{X})^\top \left(\frac{c}{c'} - 1\right)\mathbf{\Theta}_{c'}\left(\boldsymbol{\beta}(\mathbf{T}) - e^p(\mathbf{X})\right)\right\}\right.
$$
$$
\left. \cdot \left\{(f_{\mathbf{\Theta}_{c'}}(\mathbf{X},\mathbf{T}) - f^*(\mathbf{X},\mathbf{T})) - \mathbb{E}\left[f_{\mathbf{\Theta}_{c'}}(\mathbf{X},\mathbf{T}) - f^*(\mathbf{X},\mathbf{T}) \mid \mathbf{X}\right]\right\}\right] \tag{80}
$$

Denote the two terms $E_1$ and $E_2$. By the same argument as Lemma 7, where the Lebesgue-measure-non-zero condition is satisfied by Assumption 5, there exist a constant $const. > 0$ such that $E_1 \geq \left(\frac{c}{c'} - 1\right)^2 const. \|f_{\mathbf{\Theta}_{c'}}\|_{L_2} \to const. \|f^*\|_{L_2}$ as $c' \to \infty$. But for $E_2$, note that $\|f_{\mathbf{\Theta}_{c'}} - f^*\|_{L_2} \to 0$ as $c' \to \infty$. So $E_2 = o(E_1)$, and under mild conditions there exists a constant $F > 0$ such that for all $c, c'$,

$$
R\left(\frac{c}{c'}\mathbf{\Theta}_{c'}\right) - R(\mathbf{\Theta}_{c'}) \leq F\mathbb{E}\left[\left\{\boldsymbol{\alpha}(\mathbf{X})^\top \mathbf{\Theta}_{c'}\left(\boldsymbol{\beta}(\mathbf{T}) - e^p(\mathbf{X})\right)\right\}^2\right] \tag{81}
$$

. Then note:

$$\mathbb{E}\left[\left\{\boldsymbol{\alpha}(\mathbf{X})^\top \boldsymbol{\Theta}_{c'}\left(\boldsymbol{\beta}(\mathbf{T}) - e^p(\mathbf{X})\right)\right\}^2\right]$$

$$= \mathbb{E}\left[\langle \boldsymbol{\Theta}_{c'}, \boldsymbol{\alpha}(\mathbf{T}) \otimes (\boldsymbol{\beta}(\mathbf{T}) - e^p(\mathbf{X})))\rangle^2\right] \tag{82}$$

$$= \mathbb{E}\left[\langle \boldsymbol{\Theta}_{c'} \otimes \boldsymbol{\Theta}_{c'}, (\boldsymbol{\alpha}(\mathbf{T}) \otimes (\boldsymbol{\beta}(\mathbf{T}) - e^p(\mathbf{X}))) \otimes (\boldsymbol{\alpha}(\mathbf{T}) \otimes (\boldsymbol{\beta}(\mathbf{T}) - e^p(\mathbf{X})))\rangle\right] \tag{83}$$

$$= \langle \boldsymbol{\Theta}_{c'} \otimes \boldsymbol{\Theta}_{c'}, \mathbb{E}\left[(\boldsymbol{\alpha}(\mathbf{T}) \otimes (\boldsymbol{\beta}(\mathbf{T}) - e^p(\mathbf{X}))) \otimes (\boldsymbol{\alpha}(\mathbf{T}) \otimes (\boldsymbol{\beta}(\mathbf{T}) - e^p(\mathbf{X})))\right]\rangle \tag{84}$$

$$\leq \|\boldsymbol{\Theta}_{c'} \otimes \boldsymbol{\Theta}_{c'}\|\left\|\mathbb{E}\left[(\boldsymbol{\alpha}(\mathbf{T}) \otimes (\boldsymbol{\beta}(\mathbf{T}) - e^p(\mathbf{X}))) \otimes (\boldsymbol{\alpha}(\mathbf{T}) \otimes (\boldsymbol{\beta}(\mathbf{T}) - e^p(\mathbf{X})))\right]\right\| \tag{85}$$

$$= \|\boldsymbol{\Theta}_{c'}\|^2 \left\|\underbrace{\mathbb{E}\left[(\boldsymbol{\alpha}(\mathbf{T}) \otimes (\boldsymbol{\beta}(\mathbf{T}) - e^p(\mathbf{X}))) \otimes (\boldsymbol{\alpha}(\mathbf{T}) \otimes (\boldsymbol{\beta}(\mathbf{T}) - e^p(\mathbf{X})))\right]}_{constant}\right\| \tag{86}$$

$$= const.\|f_{\boldsymbol{\Theta}_{c'}}\|^2_{\mathcal{H}_{c'}} \tag{87}$$

$$= const.\|f_{\boldsymbol{\Theta}_{c'}}\|^2_{L_2} \tag{88}$$

where Eq. 85 is by Cauchy-Schwarz and the equation 86 uses the fact that under Euclidean norms for finite dimensional real vectors $\mathbf{a}, \mathbf{b}$, $\|\mathbf{a} \otimes \mathbf{b}\| = \|\mathbf{a}\|\|\mathbf{b}\|$. equation 87 is due to the vector 2-norm of $\boldsymbol{\Theta}$ is equal to the RKHS norm of $f_{\boldsymbol{\Theta}}$, and equation 88 is due to the fact that in finite dimensions all norms are Lipschitz equivalent. Note that the constant factors in 87 and 88 may be different but that both positive.

Then finally by the triangle inequality,

$$\|f_{\Theta_c} - f_{\boldsymbol{\Theta}_{c'}}\|_{L_2} \leq \|f_{\boldsymbol{\Theta}_{c'}} - \frac{c}{c'}f_{\boldsymbol{\Theta}_{c'}}\|_{L_2} + \|f_{\Theta_c} - \frac{c}{c'}f_{\boldsymbol{\Theta}_{c'}}\|_{L_2} \tag{89}$$

$$\leq \left(1 - \frac{c}{c'}\right)\|f_{\boldsymbol{\Theta}_{c'}}\|_{L_2} + constant.\|f_{\boldsymbol{\Theta}_{c'}}\|_{L_2} \tag{90}$$

$$\leq const.\|f_{\boldsymbol{\Theta}_{c'}}\|_{L_2} \tag{91}$$

again for a positive constant factor in the last equality. $\qquad\square$

Now we have arrived at the position to bound the difference between the oracle and feasible regrets by functions of the true regret. We first present Lemma 11 which bounds the difference between $\hat{R}_n$ and $\tilde{R}_n$ in terms of $R$. Then, we leverage the result by [42] to linearize the dependence on $R$.

**Lemma 11.** *Suppose that the propensity estimate $e^p(\mathbf{x})$ is uniformly consistent,*

$$\sup_{\mathbf{x} \in \mathcal{X}} \|\hat{e}^p(\mathbf{x}) - e^p(\mathbf{x})\| \to_p 0 \tag{92}$$

*and the $L_2$ errors converge at rate*

$$\mathbb{E}\left[\{\hat{m}(\mathbf{X}) - m^*(\mathbf{X})\}^2\right], \mathbb{E}\left[\|\hat{e}^p(\mathbf{X}) - e^p(\mathbf{X})\|^2\right] = \mathcal{O}(a_n^2) \tag{93}$$

*for some sequence $a_n \to 0$. Suppose, moreover, Assumptions 5, 6 and 4 hold. Then, for any $\epsilon > 0$, there exists a constant $U(\epsilon)$ such that the regret functions induced by the oracle learner and the feasible learner are coupled with probability at least $1 - \epsilon$ as*

$$\left|\hat{R}_n(\boldsymbol{\Theta}; c) - \tilde{R}_n(\boldsymbol{\Theta}; c)\right| \leq U(\varepsilon)\left\{c^p R(\boldsymbol{\Theta}; c)^{(1-p)/2}a_n^2 + c^{2p}R(\boldsymbol{\Theta}; c)^{1-p}\frac{1}{\sqrt{n}}\log(n)\right.$$

$$+ c^{2p}R(\boldsymbol{\Theta}; c)^{1-p}\frac{1}{n}\log\left(\frac{cn^{1/(1-p)}}{R(\boldsymbol{\Theta}; c)}\right) + c^p R(\boldsymbol{\Theta}; c)^{1-\frac{p}{2}}\frac{1}{\sqrt{n}}\sqrt{\log\left(\frac{cn^{1/(1-p)}}{R(\boldsymbol{\Theta}; c)}\right)}$$

$$\left. + c^p R(\boldsymbol{\Theta}; c)^{(1-p)/2}a_n\frac{1}{\sqrt{n}}\sqrt{\log\left(\frac{cn^{1/(1-p)}}{R(\boldsymbol{\Theta}; c)}\right)} + \xi_n R(\boldsymbol{\Theta}; c)\right\} \tag{94}$$

*simultaneously for all $1 \leq c \leq \log(n)$.*

*Proof.* Following [42], we start by decomposing the feasible loss function $\hat{L}_n(\boldsymbol{\Theta})$ into the oracle loss together with additional terms as follows:

$$\hat{L}_n(\boldsymbol{\Theta}) = \frac{1}{n}\sum_{l=1}^{n}\left((Y_l - \widehat{m}_{(-q(l))}(\mathbf{X}_l)) - \boldsymbol{\alpha}(\mathbf{X}_l)^\top\boldsymbol{\Theta}(\boldsymbol{\beta}(\mathbf{T}_l) - \widehat{e}^p_{(-q(l))}(\mathbf{X}_l))\right)^2 \tag{95}$$

$$= \frac{1}{n}\sum_{l=1}^{n}\Big[(Y_l - m^*(\mathbf{X}_l)) + \{m^*(\mathbf{X}_l) - \widehat{m}(\mathbf{X}_l)\} - \boldsymbol{\alpha}(\mathbf{X}_l)^\top\boldsymbol{\Theta}(\boldsymbol{\beta}(\mathbf{T}_l) - e^p(\mathbf{X}_l))$$

$$- \boldsymbol{\alpha}(\mathbf{X}_l)^\top\boldsymbol{\Theta}(e^p(\mathbf{X}_l) - \widehat{e}^p_{(-q(l))}(\mathbf{X}_l))\Big]^2 \tag{96}$$

$$= \frac{1}{n}\sum_{l=1}^{n}\Big[\{Y_l - m^*(\mathbf{X}_l)\} - \boldsymbol{\alpha}(\mathbf{X}_l)^\top\boldsymbol{\Theta}(\boldsymbol{\beta}(\mathbf{T}_l) - e^p(\mathbf{X}_l))\Big]^2$$

$$+ \frac{1}{n}\sum_{l=1}^{n}[\{m^*(\mathbf{X}_l) - \widehat{m}(\mathbf{X}_l)\} - \boldsymbol{\alpha}(\mathbf{X}_l)^\top\boldsymbol{\Theta}(e^p(\mathbf{X}_l) - \widehat{e}^p_{(-q(l))})]^2$$

$$+ \frac{2}{n}\sum_{l=1}^{n}\Big[\{Y_l - m^*(\mathbf{X}_l)\} - \boldsymbol{\alpha}(\mathbf{X}_l)^\top\boldsymbol{\Theta}(\boldsymbol{\beta}(\mathbf{T}_l) - e^p(\mathbf{X}_l))\Big]$$

$$\cdot\Big[\{m^*(\mathbf{X}_l) - \widehat{m}_{(-q(l))}(\mathbf{X}_l)\} - \boldsymbol{\alpha}(\mathbf{X}_l)^\top\boldsymbol{\Theta}(e^p(\mathbf{X}_l) - \widehat{e}^p_{(-q(l))}(\mathbf{X}_l))\Big] \tag{97}$$

$$= \frac{1}{n}\sum_{l=1}^{n}\Big[\{Y_l - m^*(\mathbf{X}_l)\} - \boldsymbol{\alpha}(\mathbf{X}_l)^\top\boldsymbol{\Theta}(\boldsymbol{\beta}(\mathbf{T}_l) - e^p(\mathbf{X}_l))\Big]^2$$

$$+ \frac{1}{n}\sum_{l=1}^{n}\Big[\{m^*(\mathbf{X}_l) - \widehat{m}_{(-q(l))}(\mathbf{X}_l)\} - \boldsymbol{\alpha}(\mathbf{X}_l)^\top\boldsymbol{\Theta}(e^p(\mathbf{X}_l) - \widehat{e}^p_{(-q(l))}(\mathbf{X}_l))\Big]^2$$

$$- \frac{2}{n}\sum_{l=1}^{n}\{Y_l - m^*(\mathbf{X}_l)\}\boldsymbol{\alpha}(\mathbf{X}_l)\boldsymbol{\Theta}(e^p(\mathbf{X}_l) - \widehat{e}^p_{(-q(l))}(\mathbf{X}_l))$$

$$- \frac{2}{n}\sum_{l=1}^{n}\boldsymbol{\alpha}(\mathbf{X}_l)^\top\boldsymbol{\Theta}(\boldsymbol{\beta}(\mathbf{T}_l) - e^p(\mathbf{X}_l))\{m^*(\mathbf{X}_l) - \widehat{m}_{(-q(l))}(\mathbf{X}_l)\}$$

$$+ \frac{2}{n}\sum_{l=1}^{n}\boldsymbol{\alpha}(\mathbf{X}_l)^\top\boldsymbol{\Theta}(\boldsymbol{\beta}(\mathbf{T}_l) - e^p(\mathbf{X}_l))\boldsymbol{\alpha}(\mathbf{X}_l)^\top\boldsymbol{\Theta}(e^p(\mathbf{X}_l) - \widehat{e}^p_{(-q(l))}(\mathbf{X}_l)) \tag{98}$$

Furthermore, we may verify that some terms cancel out when we restrict our attention to the main objective of interest

$$\widehat{R}(\boldsymbol{\Theta};c) - \tilde{R}(\boldsymbol{\Theta};c) = \hat{L}_n(\boldsymbol{\Theta}) - \hat{L}_n(\boldsymbol{\Theta}_c) - \tilde{L}_n(\boldsymbol{\Theta}) + \tilde{L}_n(\boldsymbol{\Theta}_c) \tag{99}$$

In particular, note that the first term in the decomposition above is exactly $\tilde{L}_n(\boldsymbol{\Theta})$. Thus

$$
\begin{aligned}
&\widehat{R}(\boldsymbol{\Theta};c) - \tilde{R}(\boldsymbol{\Theta};c) \\
&= -\frac{2}{n}\sum_{l=1}^{n}\{m^*(\mathbf{X}_l) - \widehat{m}_{(-q(l))}(\mathbf{X}_l)\}\boldsymbol{\alpha}(\mathbf{X}_l)^\top(\boldsymbol{\Theta}-\boldsymbol{\Theta}_c)(e^p(\mathbf{X}_l) - \widehat{e}^p_{(-q(l))}(\mathbf{X}_l)) \\
&\quad + \frac{1}{n}\sum_{l=1}^{n}\{\boldsymbol{\alpha}(\mathbf{X}_l)^\top\boldsymbol{\Theta}(e^p(\mathbf{X}_l) - \widehat{e}^p_{(-q(l))}(\mathbf{X}_l))\}^2 - \{\boldsymbol{\alpha}(\mathbf{X}_l)^\top\boldsymbol{\Theta}_c(e^p(\mathbf{X}_l) - \widehat{e}^p_{(-q(l))}(\mathbf{X}_l))\}^2 \\
&\quad - \frac{2}{n}\sum_{l=1}^{n}\{Y_l - m^*(\mathbf{X}_l)\}\boldsymbol{\alpha}(\mathbf{X}_l)^\top(\boldsymbol{\Theta}-\boldsymbol{\Theta}_c)(e^p(\mathbf{X}_l) - \widehat{e}^p_{(-q(l))}(\mathbf{X}_l)) \\
&\quad - \frac{2}{n}\sum_{l=1}^{n}\boldsymbol{\alpha}(\mathbf{X}_l)^\top(\boldsymbol{\Theta}-\boldsymbol{\Theta}_c)(\boldsymbol{\beta}(\mathbf{T}_l) - e^p(\mathbf{X}_l))\{m^*(\mathbf{X}_l) - \widehat{m}_{(-q(l))}(\mathbf{X}_l)\} \\
&\quad + \frac{2}{n}\sum_{l=1}^{n}\boldsymbol{\alpha}(\mathbf{X}_l)^\top\boldsymbol{\Theta}(\boldsymbol{\beta}(\mathbf{T}_l) - e^p(\mathbf{X}_l))\boldsymbol{\alpha}(\mathbf{X}_l)^\top\boldsymbol{\Theta}(e^p(\mathbf{X}_l) - \widehat{e}^p_{(-q(l))}(\mathbf{X}_l)) \\
&\quad - \frac{2}{n}\sum_{l=1}^{n}\boldsymbol{\alpha}(\mathbf{X}_l)^\top\boldsymbol{\Theta}_c(\boldsymbol{\beta}(\mathbf{T}_l) - e^p(\mathbf{X}_l))\boldsymbol{\alpha}(\mathbf{X}_l)^\top\boldsymbol{\Theta}_c(e^p(\mathbf{X}_l) - \widehat{e}^p_{(-q(l))}(\mathbf{X}_l))
\end{aligned}
$$
$$(100)$$

Letting $A_1^c(\boldsymbol{\Theta})$, $A_2^c(\boldsymbol{\Theta})$, $B_1^c(\boldsymbol{\Theta})$ and $B_3^c(\boldsymbol{\Theta})$ denote these 5 summands respectively, we seek to bound each of the terms in terms of $R(\boldsymbol{\Theta};c)$. Starting with $A_1^c(\boldsymbol{\Theta})$, we extract $\boldsymbol{\Theta}-\boldsymbol{\Theta}_c$ by its infinity norm and by Cauchy-Schwarz,

$$
|A_1^c(\boldsymbol{\Theta})| \le 2\sqrt{\frac{1}{n}\sum_{l=1}^{n}\left\{m^*(\mathbf{X}) - \widehat{m}_{(-q(l))}(\mathbf{X}_l)\right\}^2}
$$
$$
\cdot \sqrt{\frac{1}{n}\sum_{l=1}^{n}\left\|\boldsymbol{\alpha}(\mathbf{X}_l)\otimes\left(e^p(\mathbf{X}) - \widehat{e}^p_{(-q(l))}(\mathbf{X}_l)\right)\right\|^2} \cdot \|\boldsymbol{\Theta}-\boldsymbol{\Theta}_c\|_\infty \quad (101)
$$
$$(102)$$

Using the fact that $\|\mathbf{a}\otimes\mathbf{b}\| = \|\mathbf{a}\|\|\mathbf{b}\|$ for $\mathbf{a}$ and $\mathbf{b}$ some (finite dimensional) vector, we may separate out the norm of $\boldsymbol{\alpha}(\mathbf{X})$ and we know $\|\boldsymbol{\alpha}(\mathbf{X})\|^2$ is uniformly bounded by Assumption 6. By equation 93 and Markov's inequality, the mean squared errors of the $m-$ and $e-$models decay at rate $O_P(a_n)$. Therefore, applying 71 to bound the infinity-norm discrepancy $\|\boldsymbol{\Theta}-\boldsymbol{\Theta}_c\|_\infty$, we find that simultaneously for all $c \ge 1$,

$$
\sup\{c^{-p}R(\boldsymbol{\Theta};c)^{-\frac{1-p}{2}}|A_1^c(\boldsymbol{\Theta})| : f_{\boldsymbol{\Theta}}\in\mathcal{H}_c, c\ge 1\} = O_P(a_n^2) \quad (103)
$$

Following [42] and using a similar argument to extract $\|\boldsymbol{\alpha}(\mathbf{X})\|$ and bound $\boldsymbol{\Theta}-\boldsymbol{\Theta}_c$ by the c-regret $\|R(\boldsymbol{\Theta};c)\|$, we get that

$$
|A_2^c| = \mathcal{O}_P\left(\left(c^p R(\boldsymbol{\Theta};c)^{\frac{1-p}{2}} + c^{2p}R(\boldsymbol{\Theta};c)^{1-p}\right)a_n^2\right) \quad (104)
$$

In order to bound $B_1^c(\boldsymbol{\Theta})$, decomposing it with respect to the cross fitting structure, we consider

$$
B_{1,q}^c(\boldsymbol{\Theta}) = \frac{\sum_{\{l:q(l)\}}2\{Y - m^*(\mathbf{X})\}\boldsymbol{\alpha}(\mathbf{X}_l)^\top(\boldsymbol{\Theta}-\boldsymbol{\Theta}_c)(e^p(\mathbf{X}_l) - \widehat{e}^p_{(-q(l))}(\mathbf{X}_l))}{|\{l : q(l) = q\}|}, \quad (105)
$$

noting that $|B_1^c(\boldsymbol{\Theta})| \le \sigma_{q=1}^Q|B_{1,q}^c(\boldsymbol{\Theta})|$. In particular, we bound its supremum $\sup B_{1,q}^c(\boldsymbol{\Theta})$. To proceed, we bound this quantity over sets indexed by $c$ and $\delta$ such that $\|f_{\boldsymbol{\Theta}} - f_{\boldsymbol{\Theta}_c}\|_{L^2} \le \delta$:

$$
\sup_{\boldsymbol{\Theta}\in\mathcal{H}_c}\left\{B_{1,q}^c(\boldsymbol{\Theta}) : \|f_{\boldsymbol{\Theta}} - f_{\boldsymbol{\Theta}_c}\|_{L^2} \le \delta\right\}. \quad (106)
$$

Letting $\mathcal{I}^{(-q)} = \{\mathbf{X}_l, \mathbf{T}_l, Y_l : q(l) \neq q\}$ denote the set of data points excluded in the $q-$fold, using a similar procedure to [42], we can check that the conditional expectation $\mathbb{E}\left[B_{1,q}^c \mid \mathcal{I}^{(-q)}\right] = 0$. By conditioning on $\mathcal{I}^{(-q)}$, the summands in $B_{1,q}^c(\mathbf{\Theta})$ become independent, as $\widehat{e}^p(\mathbf{X})(\mathbf{X}_l)$ is now only random in $\mathbf{X}$.

Now, the next step in [42] is to bound the expectation of the supremum of $B_{1,q}^c$ using [42, Lemma 5] and [42, Eq. (36)]. Since we work with a vector of propensity features instead of a single propensity score unlike in [42], we need to apply [42, Lemma 5] $d$ times where $d$ is the dimension of $e^p(\mathbf{X})$:

$$
\begin{aligned}
B_{1,q}^c(\mathbf{\Theta}) &= \frac{\langle (\mathbf{\Theta} - \mathbf{\Theta}_c), \sum_{\{l:q(l)\}} 2\{Y - m^*(\mathbf{X})\}\boldsymbol{\alpha}(\mathbf{X}_l) \otimes (e^p(\mathbf{X}_l) - \widehat{e}_{(-q(l))}^p(\mathbf{X}_l)))\rangle}{|\{l : q(l) = q\}|} \quad (107) \\
&= \frac{\sum_{ij}(\mathbf{\Theta} - \mathbf{\Theta}_c)_{ij}, \sum_{\{l:q(l)\}} 2\{Y - m^*(\mathbf{X})\}\alpha_i(\mathbf{X}_l)(e_j^p(\mathbf{X}_l) - \hat{e}_{(-q(l)),j}^p(\mathbf{X}_l))}{|\{l : q(l) = q\}|}, \quad (108)
\end{aligned}
$$

so

$$
\sup_{f_{\mathbf{\Theta}} \in \mathcal{H}_c} \{B_{1,q}^c(\mathbf{\Theta})\} \leq \frac{\sum_{ij} \sup_{f_{\mathbf{\Theta}} \in \mathcal{H}_c} \sum_{\{l:q(l)\}} 2\{Y - m^*(\mathbf{X})\}\alpha_i(\mathbf{X}_l)(e_j^p(\mathbf{X}_l) - \hat{e}_{(-q(l)),j}^p(\mathbf{X}_l))(\mathbf{\Theta} - \mathbf{\Theta}_c)_{ij}}{|\{l : q(l) = q\}|}
$$
$$(109)$$

So bounding each term indexed by $ij$ using Lemma 5 of [42] and equation 93, we will get the same bound as in [42] because the sum over $ij$ is finite and $\boldsymbol{\alpha}$ is bounded.

Then, using a similar argument to [42], we may obtain that for any fixed $c, \delta, \epsilon > 0$, there exists a different constant B such that with probability at least $1 - \epsilon$,

$$
\begin{aligned}
&\sup_{\tau \in \mathcal{H}_c} \left\{ B_{1,q}^c(\tau) \mid \mathcal{I}^{(-q)} : \|f_{\mathbf{\Theta}} - f_{\mathbf{\Theta}_c}\|_{L_2} \leq \delta \right\} \\
&\quad < B \left\{ c^p \delta^{1-p} a_n \frac{\log(n)}{\sqrt{n}} + \frac{c^p \delta^{1-p} a_n}{\sqrt{n}} \sqrt{\log\left(\frac{1}{\varepsilon}\right)} + \frac{1}{n} c^p \delta^{1-p} \log\left(\frac{1}{\varepsilon}\right) \right\},
\end{aligned} \quad (110)
$$

which holds unconditionally of $\mathcal{I}^{(-q)}$. In order to establish the bound for all values of $c$ and $\delta$ simultaneously, we may proceed with the same argument as [42]; instead of [42, Lemma 6], we replace with our Lemma 10, which is our extension to the multidimensional setting. $B_2^c(\mathbf{\Theta})$ may be bounded similarly.

To bound $B_3(\mathbf{\Theta})$, the argument of [42] is easily extended as well, using the decomposition which we detail below.

To simplify notation, write

$$
\mathbf{a}_l = \boldsymbol{\alpha}(\mathbf{X}_l) \otimes \left(\boldsymbol{\beta}(\mathbf{T}_l) - e^p(\mathbf{X}_l)\right) \quad (111)
$$

$$
\mathbf{b}_l = \boldsymbol{\alpha}(\mathbf{X}_l) \otimes \left(e^p(\mathbf{X}_l) - \widehat{e}_{(-q(l))}^p(\mathbf{X}_l)\right) \quad (112)
$$

Note:

$$B_3^c = \frac{2}{n}\sum_{l=1}^n \langle \boldsymbol{\Theta}, \mathbf{a}_l\rangle\langle \boldsymbol{\Theta}, \mathbf{b}_l\rangle - \frac{2}{n}\sum_{l=1}^n \langle \boldsymbol{\Theta}_c, \mathbf{a}_l\rangle\langle \boldsymbol{\Theta}_c, \mathbf{b}_l\rangle \tag{113}$$

$$= \frac{2}{n}\sum_{l=1}^n \bigg\{ 2\langle \boldsymbol{\Theta}, \mathbf{a}_l\rangle\langle \boldsymbol{\Theta}, \mathbf{b}_l\rangle - \langle \boldsymbol{\Theta}, \mathbf{a}_l\rangle\langle \boldsymbol{\Theta}, \mathbf{b}_l\rangle$$

$$- \langle \boldsymbol{\Theta}_c, \mathbf{a}_l\rangle\langle \boldsymbol{\Theta}, \mathbf{b}_l\rangle + \langle \boldsymbol{\Theta}_c, \mathbf{a}_l\rangle\langle \boldsymbol{\Theta}, \mathbf{b}_l\rangle$$
$$- \langle \boldsymbol{\Theta}, \mathbf{a}_l\rangle\langle \boldsymbol{\Theta}_c, \mathbf{b}_l\rangle + \langle \boldsymbol{\Theta}, \mathbf{a}_l\rangle\langle \boldsymbol{\Theta}_c, \mathbf{b}_l\rangle$$

$$- \langle \boldsymbol{\Theta}_c, \mathbf{a}_l\rangle\langle \boldsymbol{\Theta}_c, \mathbf{b}_l\rangle \bigg\} \tag{114}$$

$$= \frac{2}{n}\sum_{l=1}^n \bigg\{ \langle \boldsymbol{\Theta} - \boldsymbol{\Theta}_c, \mathbf{a}_l\rangle\langle \boldsymbol{\Theta}, \mathbf{b}_l\rangle + \langle \boldsymbol{\Theta}, \mathbf{a}_l\rangle\langle \boldsymbol{\Theta} - \boldsymbol{\Theta}_c, \mathbf{b}_l\rangle \tag{115}$$

$$- \langle \boldsymbol{\Theta} - \boldsymbol{\Theta}_c, \mathbf{a}_l\rangle\langle \boldsymbol{\Theta} - \boldsymbol{\Theta}_c, \mathbf{b}_l\rangle \bigg\} \tag{116}$$

$$\leq \left| \frac{2}{n}\sum_{l=1}^n \langle \boldsymbol{\Theta} - \boldsymbol{\Theta}_c, \mathbf{a}_l\rangle\langle \boldsymbol{\Theta}, \mathbf{b}_l\rangle \right|$$

$$+ \left| \frac{2}{n}\sum_{l=1}^n \langle \boldsymbol{\Theta}, \mathbf{a}_l\rangle\langle \boldsymbol{\Theta} - \boldsymbol{\Theta}_c, \mathbf{b}_l\rangle \right|$$

$$+ \frac{2}{n}\sum_{l=1}^n \|\boldsymbol{\Theta} - \boldsymbol{\Theta}_c\|_2^2 \|\mathbf{a}_l\|_2 \|\mathbf{b}_l\|_2 \tag{117}$$

where the last term of the last inequality follows by Cauchy-Schwarz.

The first two terms can be bounded similarly to the argument used for bounding $B_1^c(\boldsymbol{\Theta})$. For the last term, we note that $\|\boldsymbol{\Theta} - \boldsymbol{\Theta}_c\|_2 = \|f_{\boldsymbol{\Theta}} - f_{\boldsymbol{\Theta}_c}\|_{L_2}$ since by construction the RKHS norm and the $L_2$ norms are equal. Therefore, the last term is bounded by $\xi_n \|f_{\boldsymbol{\Theta}} - f_{\boldsymbol{\Theta}_c}\|_{L_2}$ where

$$\xi_n = \|\boldsymbol{\alpha}(\mathbf{X}_l) \otimes (\boldsymbol{\beta}(\mathbf{T}_l) - e^p(\mathbf{X}_l))\|_\infty \|\boldsymbol{\alpha}(\mathbf{X}_l) \otimes (e^p(\mathbf{X}_l) - \widehat{e}^p_{(-q(l))}(\mathbf{X}_l))\|_\infty = o(1). \tag{118}$$

Note that we do not need the lower order terms present in [42] which followed from [42, Lemma 7].

Thus the desired result follows. □

By [42, Lemma 2], Lemma 11 implies that under Assumptions 6 to 4, and the conditions in Lemma 11 and Lemma 4, where the $(a_n)$ in Lemma 11 is such that $a_n = O(n^{-\kappa})$ with $\kappa > \frac{1}{4}$, then

$$\left| \hat{R}_n(\boldsymbol{\Theta}; c) - \tilde{R}_n(\boldsymbol{\Theta}; c) \right| \leq 0.125 R(\boldsymbol{\Theta}; c) + o(\rho_n(c)) \tag{119}$$

with probability at least $1 - \epsilon$, for all $\boldsymbol{\Theta} \in \mathcal{H}_c$, $1 \leq c \leq \log(n)$ for large enough n.

Thus we have finally bridged $\hat{R}_n$ and $\tilde{R}_n$ with respect to the expected regret $R$. We are ready to prove our main theorem which concerns the regret bound of $\hat{R}_n$.

### F.8  Using the bridge result to derive feasible regret bound

**Theorem 2.** *Under Assumptions 5, 6, 4 and the conditions in Lemma 11 and Lemma 4, where the $(a_n)$ in Lemma 11 is such that $a_n = O(n^{-\kappa})$ with $\kappa > \frac{1}{4}$, and suppose that we obtain $\hat{\boldsymbol{\Theta}}$ via a penalized basis function regression variant of the generalized R-learner, with a properly chosen penalty of the form $\Lambda_n(\|\hat{\boldsymbol{\Theta}}\|_2)$ that grows faster than $\rho_n(\|\hat{\boldsymbol{\Theta}}\|_2)$ in 53 . Then $\hat{\boldsymbol{\Theta}}$ satisfies the same regret bound as $\tilde{\boldsymbol{\Theta}}$, $R(\hat{\boldsymbol{\Theta}}_n) = \tilde{O}(n^{\frac{1}{1+p}})$.*

*Proof.* We have established that when we set $\rho_n$ as

$$\rho_n(c) = U(\epsilon)\{1 + \log(n) + \log\log(c+e)\}\left(\frac{(c+1)^p \log(n)}{\sqrt{n}}\right)^{2/(1+p)},$$

we have that for every $\epsilon$ there exist a constant $U(\epsilon)$ such that for large enough $n$ the following is satisfied with probability at least $1 - \epsilon$:

$$\frac{1}{2}\tilde{R}_n(f_\Theta; c) - \rho_n(c) \le R_n(f_\Theta; c) \le 2\tilde{R}_n(f_\Theta; c) + \rho_n(c) \tag{120}$$

Subsection F.6 argued that this leads to a rate of $\tilde{\mathcal{O}}(n^{-\frac{1}{1+p}})$ for $R(\tilde{\Theta})$.

Now to show that feasible learner matches the rate of the oracle learner,

Eq. 119 implies that

$$R(\Theta; c) \le 2\tilde{R}_n(\Theta; c) + \rho_n(c) \tag{121}$$
$$\le 2\hat{R}_n(\tau; c) + 0.25kR(\tau; c) + k\rho_n(c) \tag{122}$$

Rearranging the inequality implies that

$$R(\Theta; c) \le \frac{8}{3}\hat{R}_n(\Theta; c) + 2\rho_n(c) \tag{123}$$

for large $n$ for all $1 < c < \log(n)$, with probability at least $1 - 2\epsilon$. It can then be checked following a symmetrical argument, that

$$\frac{3}{8}\hat{R}_n(\Theta; c) - 2\rho_n(c) \le R(\Theta; c) \le \frac{8}{3}\hat{R}_n(\Theta; c) + 2\rho_n(c) \tag{124}$$

for $n$ large enough for all $1 \le c \le \log(n)$ with probability at least $1 - 4\epsilon$.

Then, following the same argument as [42], we find that the feasible minimizer has the same regret bound as the oracle minimizer: $R(\hat{\Theta}_n) = \tilde{\mathcal{O}}\left(n^{-\frac{1}{1+p}}\right)$.

This is to say:

$$\boxed{R(\hat{\Theta}_n) = O(r_n^2), \; r_n = n^{-\frac{1}{2(1+p)}}} \tag{125}$$

$\square$