# OpenReview forum: "Causal Effect Inference for Structured Treatments"
_NeurIPS.cc/2021/Conference — NeurIPS 2021 Poster_

### Official Review · Reviewer_5P78 · 2021-07-13

**Rating:** 6
**Confidence:** 4

**Summary:**

This paper proposed a CATE estimation approach when the treatment variable $T$ is a graph. Under a separable assumption (A3) between confounders $X$ and the treatment $T$ in the outcome structural equation ($Y$), Robinson's decomposition of CATE for binary $T$ can be applied. The performance of the authors' proposal was evaluated in both synthetic experiments and real data analysis.

**Ethical Concerns:**

No ethical concerns.

**Limitations And Societal Impact:**

Yes. The authors listed a few limitations and what can be done in future work and wrote a quite thorough paragraph on the potential negative societal impact of applying causal inference methods in real life.

**Main Review:**

The authors set out to address a very important problem in causal inference, which is methodology development for non-Euclidean treatment variables. In particular, the authors considered graph-valued treatment $T$. However, from my own perspective, there are several major limitations in this work.

(1) The type of intervention on graph-valued treatment is not spelled out clearly in the paper (or at least in the introduction section).

As a referee, when I saw the title of this paper, the immediate question I had in mind is what type of intervention on graphs the authors were considering. But until Section 4, I finally realized that the intervention is simply changing one graph $\mathcal{G}$ to another $\mathcal{G}'$. I think this should be made much clearer at the beginning of the paper.

(2) As a follow-up comment of (1), I would think it will be more interesting to consider problems like the direct effect of changing one set of vertices on a graph and the indirect effect of changing this set of vertices on the overall network structure, further on to the outcome. Such effect decomposition might be more interesting to applied researchers. For example, there might be targeted cancer therapy that can inhibit the expression of one particular gene. Then it is very interesting to understand whether the effect of such a drug is manifested through its target gene or through the entire gene regulatory networks. Directly comparing the change of the network may not be entirely meaningful to applied researchers.

(3) Kennedy (2020) proposed a novel CATE estimation method based on the efficient influence function. I am wondering if this or related method can also be used in the graph-valued treatment setup.

Some of the following minor limitations might have been pointed out by the authors themselves. If so, I will not describe in more detail:
(1) Lack of theoretical guarantees. But personally, I do not think this is a very important limitation for a really novel idea.
(2) In Section 4.4, with partial data, I thought the best one can do is to produce bounds on the treatment effect. But maybe I am wrong.

References:
Kennedy EH. Optimal doubly robust estimation of heterogeneous causal effects. 2020

**Time Spent Reviewing:**

2

---

> ### Author Response · Authors · 2021-08-09
> **We thank the reviewer for discussion, references and careful suggestions for future work.**
>
> We are particularly pleased that the reviewer kindly agrees that we "_address a very important problem_" with "_a really novel idea_". However, we respectfully disagree with main comment 2. We respond to each comment below.
>
> ### Main comments:
>
> 1. We tried to make it clear in Figure 1 that each treatment $\mathbf{t}$ corresponds to a different graph. We will move the definitions $\mathbf{t} \triangleq \mathcal{G}$ up near Figure 1 and Equation 1 to make it even clearer.
>
> 2. The primary motivation for our definition of graph interventions is that the nodes in the treatment graph can have constitutive, rather than causal relationships among themselves. That is, in many domains, mediation estimands resulting from node/edge changes are not meaningful. For example, removing a node from a molecule is not naturally cast as a causal effect of a node on another.
>      There is, of course, the possibility that in some domains, one may be able to partition the graph into groups of nodes for which causal relationships can be defined across groups. However, we see the problem of defining separate interventions to disjoint parts of the graph as a problem of sequential treatments, and so out of scope for this current paper.
>
> 3. We are familiar with such a family of methods, and, indeed, a version of our method constructed from influence functions is certainly interesting. The main practical issue we see is whether it will rely on estimated propensity scores $p(\mathbf T \mid \mathbf X)$, as in the dimensionality-reduction ATE estimator of [1]. This has practical implications since $\mathbf T$ can be high-dimensional, with multivariate continuous side-information on edges and nodes, and even of varying dimensionality. We need a theory that considers representation learning of low-dimensional propensity features, and principles like the one used by, e.g., Kennedy, are not there yet.
>
> ### Minor comments:
>
> 1. We have worked on some of these theoretical guarantees in the meantime and will add them to the next version. See more in the separate post titled "Quasi-Oracle Convergence Guarantee".
>
> 2. We are afraid that we do not understand this comment. We do not see why the benefit of being able to incorporate partial data into training the nuisance components should not be possible, or its relationship to bounds (which type of bounds?) on the effect.
>
>
> ### References:
>
> [1]: Nabi, Razieh, Todd McNutt, and Ilya Shpitser. "Semiparametric Causal Sufficient Dimension Reduction Of High Dimensional Treatments." arXiv preprint arXiv:1710.06727 (2017). https://arxiv.org/pdf/1710.06727.pdf

---

> > ### Comment · Reviewer_5P78 · 2021-08-11
> > **thank you for the clarification**
> >
> > I want to thank the authors for providing more clarifications and the reason why the authors consider the setting in this paper instead of the setting I was thinking about. The very reason that I thought about my proposal was due to the authors' motivating example (e.g. molecular structure of the graphs, spatial networks of regional transmissions) where causal relationships among the vertices are of interest in many contexts. I am willing to raise my score to 6 (marginally above the acceptance threshold). But I do hope that the authors can add sentences similar to "The primary motivation for our definition of graph interventions is that the nodes in the treatment graph can have constitutive, rather than causal relationships among themselves" in the camera-ready version of this paper.

---

> > > ### Author Response · Authors · 2021-08-11
> > > **Thank you very much for this feedback.**
> > >
> > > We will add the clarifying sentences as you suggested to describe the setting we have in mind more precisely. Thank you again.

---

### Official Review · Reviewer_rvHB · 2021-07-16

**Rating:** 6
**Confidence:** 4

**Summary:**

This paper studies the estimation of conditional average treatment effect (CATE) with multiple treatments that are graph structured. The proposed method extends the R-learner (Nie and Wager, 2020), which is designed for binary treatment, by learning a representation that reduces the dimension of the treatments. The representation is parametrized by a graph neural network that takes the graph structure into account. Unlike the standard R-learner that learns the nuisance components $E[Y|X]$ and $E[T|X]$ separately from the CATE, the proposed method only estimates $E[Y|X]$ separately while learns the representation $h(T)$, the propensity features $E[h(T)|X]$, and CATE jointly via gradient-based methods.

**Limitations And Societal Impact:**

(1) I would suggest stating the product effect model (9) directly without describing the general version (8), which looks odd to me. In fact, R-learner, or Robinson's method, is designed for the product effect model which breaks the response function $f(x, t)$ into the target parameter (i.e., CATE in this case) and two nuisance functions. It is shown that (e.g., Nie and Wager, 2020) R-learner yields a faster rate for the target parameter if only the two nuisance functions are estimated at an $O(n^{-1/4})$ rate. Without this structure, there is no reason to estimate $f$ through (8); instead, one can directly estimate it by minimizing $(1/N)\sum_{i=1}^{N}(y_i - f(x_i, t_i))^2$.

(2) The overlap assumption (Assumption 2) is arguably too strong when the treatment is complex. Under the model (9), is it fine to just assume overlap on $p(h(t)|x)$?

(3) Algorithm a (GIN training) updates $\eta$ only once for every $K$ updates of $\phi$ and $\eta$. Is there any specific reason for this asymmetry?

(4) It would be nice to provide a slightly more detailed description on how to parametrize the treatment via a graph neural network.

(5) An advantage of cross-fitting is to reduce bias and enable valid inference (e.g., valid pointwise confidence intervals). Is there any approach for inference with the proposed method?

**Main Review:**

The idea of combining R-learner and representation learning to reduce the dimension of complex treatments is quite interesting. The method can be useful in many applications that involve graph-structured treatments, such as those in two-sided markets. The paper is well-written overall and the technical arguments are sound.

**Time Spent Reviewing:**

4

---

> ### Author Response · Authors · 2021-08-09
> **We thank the reviewer for the insightful and encouraging comments.**
>
> 1. Yes, we agree that the current presentation of the product effect model needs a revision. We originally chose to describe the general version first to make it more familiar to readers who are already familiar with unconfoundedness, overlap, and Robinson's decomposition. However, we agree that ultimately this explanation could be rather confusing, even to readers well-versed in causality, and so we will improve the text based on your suggestions. Thanks!
>
> 2. You are right: since the initial submission we have discovered that both identifiability and convergence results indeed only depend on $p\big (h(\mathbf t) \mid g(\mathbf x)\big)$ (for more details see the comment titled 'Quasi-Oracle Convergence Guarantee').
>
> 3. Great question. There is a reason for this asymmetry: we observed that setting $K>1$, updating $\boldsymbol{\psi}, \boldsymbol{\phi}$ more frequently than $\boldsymbol{\eta}$, stabilizes the training process. Our inspiration for doing this was that we saw similar tricks used to stabilize other bi-level optimization problems including Generative Adversarial Networks.
>
> 4. We agree that a more thorough explanation of how to parameterize the GNN treatment network would be instructive, we will add this detail to the Appendix. Thank you for pointing this out.
>
> 5. Excellent question! We believe it is possible to do cross-fitting however it is a bit delicate to implement: the current training algorithm creates a dependency between learning $\widehat{e}\_{\boldsymbol{\eta}^{(i)}}^{h}(\mathbf{x})$ and data point $(\mathbf{x}\_i, \mathbf{t}\_i, y\_i)$. One way to get around this is to first learn $N$ versions of $\widehat{h}\_{\phi^{(i)}}(\mathbf{T})$, where each version omits a single training point $i$, then to learn $N$ versions of nuisance models: $\widehat{m}\_{\boldsymbol{\theta}^{(i)}}(\mathbf{X}), \widehat{e}\_{\boldsymbol{\eta}^{(i)}}^{h}(\mathbf{X})$. Finally, we learn $\widehat{g}\_{\psi}(\mathbf{X})$ on all data (using the appropriate $\widehat{h}\_{\phi^{(i)}}(\mathbf{T})$ and nuisance models). The nice thing about this is that we should avoid incurring additional variance, compared to methods that only learn one version of each model on fixed, independent splits of the data. The downside is that this approach incurs additional computation and memory. In general we find this a very interesting area of future work to develop.

---

> > ### Comment · Reviewer_rvHB · 2021-08-17
> > **Thank you for the response**
> >
> > I would like to thank the authors for their detailed response. The problem with complex treatments is important and the proposed method is promising. The additional response "Quasi-Oracle Convergence Guarantee" is convincing. I will raise my score to 7.

---

> > > ### Author Response · Authors · 2021-08-17
> > > **Thank you very much for taking the time to read our rebuttal and for your feedback.**
> > >
> > > We highly appreciate it.

---

### Official Review · Reviewer_kq9J · 2021-07-16

**Rating:** 7
**Confidence:** 5

**Summary:**

The paper considers the problem of estimating conditional average treatment effect when the treatment is a graph. The paper generalizes the Robinson decomposition, adapts the R-learner to this graph-treatment setting and introduces a two-stage algorithm. The performance of the proposed algorithm is compared to a few benchmarks through numerical experiments.

**Limitations And Societal Impact:**

The authors have adequately addressed the limitations and potential negative societal impact of their work. They have discussed limitations including lack of a convergence guarantee and potential hidden confounders.

**Main Review:**

Originality: The proposed method is original. The paper generalizes the Robinson decomposition and adapts the R-learner in a very interesting and novel way. The two-stage training algorithm is different from the cross-fitting method considered in R-learner.

Quality: As far as I checked, all claims are well supported. The experimental results are explained very clearly. One question I have is on cross-fitting. My impression is that cross-fitting is usually very helpful (sometimes even necessary) in developing theories. In practice, it is not the most data efficient way, since it uses only part of the data to train things. I’m wondering in the experiments, how much improvement in accuracy owes to not doing cross-fitting? Do you think there are specific settings where not doing cross-fitting would make the method fail? Would theories still go through without cross-fitting? I understand there is no guarantee of convergence so far, but I think a slightly deeper discussion on these would be very helpful.

Clarity: The paper is well written and I enjoyed reading it.

Significance: The problem this paper considers is very important. The paper also provides a strong solution to the problem. The proposed algorithm appears to perform very good empirically. Indeed, I believe the proposed method works for any non-binary treatment variables. It seems that both the Robinson decomposition and the two-stage algorithm can be generalized to other types of treatment, e.g., real numbers, vectors (potentially high dimensional), or categorical variables. I wonder whether this is true and how much the algorithm needs to be modified to adapt to other types of variables. If this can be discussed in depth, I think this can make the conclusions of the paper even stronger.

**Time Spent Reviewing:**

3

---

> ### Author Response · Authors · 2021-08-09
> **We thank the reviewer for the insightful and encouraging comments.**
>
> ### *I'm wondering in the experiments, how much improvement in accuracy owes to not doing cross-fitting?*
>
> Great question! It is very subtle: in the original R-learner paper [44], they train multiple, say $M$, nuisance models. Then, when they learn the causal effect for each input $i$, they use a nuisance model which has not been trained on input $i$. If we let $M$ go to $N$, where $N$ is the size of the data-set, then each nuisance model is trained on all data except one point, and we learn the causal effect using all data points and regain full efficiency. This procedure is clever as it mitigates the increased model variance from fitting nuisance/causal-effect on a significantly reduced dataset. Further, cross-fitting reduces variance (see e.g., Fig. 2 in [1]).
>
> We have not tested cross-fitting yet as it is a bit delicate to implement: the current training algorithm creates a dependency between learning  $\widehat{e}^{h}\_{\boldsymbol{\eta}^{(i)}}(\mathbf{x})$ and data point $(\mathbf{x}\_i, \mathbf{t}\_i, y\_i)$. One way to get around this is to first learn $N$ versions of $\widehat{h}\_{\phi^{(i)}}(\mathbf{T})$, where each version omits a single training point $i$, then to learn $N$ versions of nuisance models: $\widehat{m}\_{\boldsymbol{\theta}^{(i)}}(\mathbf{X}), \widehat{e}\_{\boldsymbol{\eta}^{(i)}}^{h}(\mathbf{X})$. Finally, we learn $\widehat{g}\_{\psi}(\mathbf{X})$ on all data (using the appropriate $\widehat{h}\_{\phi^{(i)}}(\mathbf{T})$ and nuisance models). We expect that this should improve the performance by reducing the bias due to not cross-fitting.
>
> ### *Do you think there are specific settings where not doing cross-fitting would make the method fail?*
>
> Also, an excellent question. In general, not cross-fitting will always increase model bias. One setting where not doing cross-fitting would have particularly damaging effects is in the large data regime. Specifically, suppose we have enough data such that cross-fitting would push the generalization error of each estimand down to a desirable level $\epsilon$ (i.e., by reducing model variance). In that case, cross-fitting will further reduce bias in CATE estimation. However, in this case, not cross-fitting would be harmful: no matter how much data we have, we would still get a biased CATE estimate.
>
> ### *Would theories still go through without cross-fitting?*
>
> Unfortunately, we do not see a route for proving these results without cross-fitting. Even extremely generic results which require nuisance estimation require cross-fitting for theoretical guarantee see, e.g., Meta-Algorithm 1 in [2]. This paper points out how cross-fitting is fundamental to semi-parametric inference in the presence of nuisance parameters.
>
> ### *Other types of treatments*:
>
> Yes, you are absolutely correct, our method works with any vectorized treatment. These vectors can result from (pre-defined or learned) feature extraction of various objects (e.g., sets, trees, images, language, graphs). We focus on graphs in this paper as we believe it makes the setting more concrete, enables exciting applications, and is overall quite general (e.g. it includes grid-like data, such as images and language, as special cases). While we briefly mention this in footnote 1, we will emphasize this more clearly in the next paper iteration. Thank you for drawing attention to this!
>
> ### References:
>
> [1]: Victor Chernozhukov, Denis Chetverikov, Mert Demirer, Esther Duflo, Christian Hansen, Whitney Newey, James Robins, Double/debiased machine learning for treatment and structural parameters, The Econometrics Journal, Volume 21, Issue 1, 1 February 2018, Pages C1–C68. https://arxiv.org/pdf/1608.00060.pdf
>
> [2]: Foster, Dylan J., and Vasilis Syrgkanis, Orthogonal statistical learning, arXiv preprint arXiv:1901.09036 (2019). https://arxiv.org/pdf/1901.09036.pdf

---

### Official Review · Reviewer_1AJd · 2021-07-16

**Rating:** 6
**Confidence:** 3

**Summary:**

This paper proposes graph intervention network (GIN) to estimate conditional average causal effects when treatments are graph-structured. GIN builds on an extension of traditional Robinson decomposition to arbitrary treatment types. In particular, GIN factorizes the outcome Y into an inner product of two terms $g(X)^\top h(T)$. During training, GIN first estimates a mean outcome model $m(X)$ and then estimate $g,h$ using the estimated $m(X)$ function. The method is evaluated on two datasets with synthetic outcome functions.

**Limitations And Societal Impact:**

The paper sufficiently addressed its limitations and societal impacts.

**Main Review:**

Overall, the proposed method is an interesting algorithmic solution to graph-structured treatment effect estimation. The method is a novel approach and the assumption on product effect is reasonable. My major concern of the paper are the following:

1. Assumption on overlap. When treatment itself is graph-structured or high-dimensional, it is hard to satisfy the overlap assumption in general. There are many unseen combinations of covariates, treatments, and outcomes $(x,t,y)$. Therefore, for many treatments $t$, the empirical estimate of $p(t|x)=0$. I don't think this paper has sufficiently addressed this issue.

2. The TCGA experimental setup is too synthetic. The outcome function is purely simulated (a predefined function). In addition, molecules from the QM9 datasets are not really drug like and they are typically not qualified as chemotherapy compounds. Using drug like molecules from the ChEMBL database is a more realistic setup.

**Time Spent Reviewing:**

1 hour

---

> ### Author Response · Authors · 2021-08-09
> **We thank the reviewer for the insightful and encouraging comments.**
>
> 1. Thank you for this question! In principle, the assumption on overlap actually only needs to hold for the true distribution $p(\mathbf T \mid \mathbf X)$, as opposed to an empirical estimate of it. Even though both $\mathbf T$ and $\mathbf X$ are high-dimensional, we believe this is a reasonable assumption in real-world scenarios. For example, if we wanted to understand the outcome $Y$ of drugs $\mathbf T$ on cancer given someone's medical record $\mathbf X$ from observational data, a sensible thing to do is only consider drugs and individuals for which it is reasonable that any drug could be taken by any individual. This implies that $0 < p(\mathbf T \mid \mathbf X) < 1$. In practice, if there are individuals or drugs which look so completely different from all other individuals or drugs then it makes sense that they should not be used as part of a causal effect estimation query. Thank you again for bringing this up, we will make this clearer in the text.
>
> 2. Yes, we agree that it would be interesting to test our method on real-world datasets with graph-structured treatments. However, it is extremely difficult to get conditional average treatment effect accuracy results from observational data alone. This is because it requires that two different treatments are applied to the same set of covariates (e.g., two different drugs are given to two identical patients), something that is extremely rare in real-life. We agree with you that the compounds in QM9 are not typical drug treatments, the reason we opted for this dataset was its ease of use and as a proof of concept. This said we will try using ChEMBL instead. Thank you for this advice!

---

### Official Review · Reviewer_fdib · 2021-07-19

**Rating:** 6
**Confidence:** 4

**Summary:**

The paper proposed a CATE estimation method for graph-structured treatment by Robinson Decomposition. It is an extension of Robinson Decomposition, and the experiment shows the effectiveness of GIN on two small world datasets. The writing is good, but some contents are missing, so that the paper is not that easy to read. My detailed comments and concerns are described in the following.

**Main Review:**

1. My biggest concern is about m(x). m(x) is related to the  e^h(x), according to the definition. However, in the training algorithm, the estimated function m(x) is fixed after Stage 1, only with the observational x and y. The design of training mismatch Eq(11), and e^h(x) is always updating when training the model. How can m(x) be fixed in the first place?

2. how to mitigate dependency between X and T towards unbiased treatment effect estimation? It is not clear in the manuscript.

3. GIN seems to work well on mitigating the selection bias according to the experiments. Which component of GIN contributes to controlling it? It is also not clear in the paper.

4. GIN uses GNN to handle the graph treatment, i.e., developing the function h(T). However, its detailed mapping process is missing. For example, how to get the readout output of the graph treatment?

5. the paper aims to estimate the CATE, so how to select the covariate values x in the training set? or use all values shown in the training set?

6. Under the setting of the GIN model, the identification of causal effect is another concern.

7. The number of baseline methods is a bit limited. Authors can use more treatment effect estimation models and expand them to the graph-treatment setting. For example, using the CFR model to obtain covariates' representation and use GNN to map the treatment into low-dimensional space, then concatenate them to predict potential outcomes.

8. BTW, as far as I know, GIN usually means "Graph Isomorphism Network", which is published in ICLR 2019.

**Time Spent Reviewing:**

4 hours

---

> ### Author Response · Authors · 2021-08-09
> **We thank the reviewer for the insightful and encouraging comments!**
>
> 1. Great question! The reason $m\left(\mathbf{x}\right)$ can be fixed is because it is defined as $\mathbb{E}\left[Y \mid \mathbf{x}\right]$. This means one can estimate this quantity once using any supervised learning method. This is subtle, as you point out $m\left(\mathbf{x}\right)$ also equals $g\left(\mathbf{x}\right)^\top e^h\left(\mathbf{x}\right)$.
>
> The motivation behind the nuisance components $m\left(\cdot\right)$ and $e^h\left(\cdot\right)$ can be understood in the simplified binary case of the R-learner [1] as follows: Eq. (4) ties together $m\left(\cdot\right)$ and $e^p(\cdot)$ $\tau_b(\cdot)$. Solving (4) directly for $\tau_b(\cdot)$ is a problem as in practice we only have estimates $\hat m\left(\cdot\right)$ and $\hat e^p\left(\cdot\right)$ -- but notice that even there we act as-if we know $m(\cdot)$ and $e^p(\cdot)$ as two functionally independent known quantities.
>
> The idea of the R-learner is to *reparameterize* the problem as Eq. (5), so that we still act as-if we know the *functionals* $m(\cdot)$ and $e^p(\cdot)$ (they are black-boxes that we can estimate with any supervised learning technique), but we use the data directly to smooth out the target parameter of interest, $\tau_b(\cdot)$. It removes the black-box $m(\cdot)$ (whose internal structure is irrelevant) from the learning problem of estimating the "_white-box_" $\tau_b(\cdot)$, which is only well-defined in the context of a causal model.
>
> Isolating these quantities is particularly advantageous, because otherwise smoothing out a parameter of interest in a sea of nuisance parameters is tricky: a nice explanation for the linear case is given in [2].
>
> Thank you for drawing attention to this, we will clarify it in the paper.
>
> 2. Our approach mitigates the dependence between $\mathbf X$ and $\mathbf T$ by generalizing the Robinson decomposition and utilizing Assumption 3 (which states that effects $Y$ are products between basis functions on $\mathbf X$ and $\mathbf T$). These allow us to directly use the arguments of the R-learner [1] and [2] that demonstrate unbiased estimation of the causal effect.
>
> 3. Thanks for this. We believe that because GIN learns propensity features $e^{h}(\mathbf{X})$, a generalization of propensity scores that reduces dimensionality while tapping into the idea of partialing nuisance parameters out of the estimation of the CATE function $g(\mathbf x)$. This can be seen in the experiments when comparing GIN to GraphITE and GNN models which do not learn to separate nuisances directly.
>
> 4. We use the PyTorch Geometric [3] implementations of k-dimensional GNNs (kGNNs) [4] for the Small-World (SW) experiment and Relational Graph Convolutional Networks (RCGNs) [5] for The Cancer Genomic Atlas (TCGA) experiment. For node aggregation, we use the default mechanisms of kGNNs (sum) and RCGNs (mean). We use BatchNorm within the GNN layers across all experiments, 3-6 GNN layers with 50-400 dimensions (depending on the experiment; chosen through random search hyper-parameter tuning). We did not detail these in the main body of the paper to emphasize that GIN allows one to _plug-in_ their favorite GNN model for their application. The hyper-parameters are in the Appendix; however, if you would like to see more details on (these) GNNs, we would be happy to add such.
>
> 5. Thanks for this inspiring question. Similar to prior CATE estimation work we assume the covariate values in the dataset have been preselected to satisfy the causal graph in Figure 1. However, we believe that learning this selection for the case of graph-structured treatments is an interesting direction for future work.
>
> 6. We are afraid we did not quite understand the question. The identification assumptions are given in lines 87-89. We agree that e.g. evaluating sensitivity to such assumptions is important, but is a topic for future work.
>
> 7. Thanks for this. We believe that extending existing treatment effect estimation models designed for binary or continuous treatments, like CFR, is actually not straightforward. There are two main reasons:
>
>     1) Estimators like CFR are trained end-to-end and use a separate neural network prediction head for each treatment level. For CFR specifically, they learn a separate covariate representation for each treatment level. However, for graph-structured treatments there can be hundreds of thousands of treatment levels, some of which may not have been seen during training. There are three downsides to this:
>         * (a) the computational and memory complexity of learning this many prediction heads is large;
>         * (b) this model cannot easily take into account the fact that certain treatments are similar (e.g., similar drugs) as each prediction head uses different weights;
>         * (c) if some treatments are not seen then there will be no gradient information used to train those prediction heads.
>
>     2) To balance representations and account for confounding, estimators like CFR rely on the notion of a pair-wise distance between control and treatment group (e.g. an integral probability metric in CFR). It is not clear how to extend such distance metrics to more than two treatment levels. Further, even if this was possible, it is still unclear how to handle unseen treatments during inference time.
>
>     To the best of our knowledge, the only method that aims to learn balanced representations across multiple treatments while not relying on a pair-wise distance between two treatment levels is GraphITE [6] which we compare against.
>
> 8. Thank you for pointing that out. We will consider changing this acronym.
>
>
> ### References:
>
> * [1]: X Nie, S Wager, Quasi-oracle estimation of heterogeneous treatment effects, Biometrika, Volume 108, Issue 2, June 2021, Pages 299–319, https://doi.org/10.1093/biomet/asaa076
> * [2]: P. Richard Hahn, Carlos M. Carvalho, David Puelz, Jingyu He, Regularization and Confounding in Linear Regression for Treatment Effect Estimation, Bayesian Analysis, Bayesian Anal. 13(1), 163-182, (March 2018). https://arxiv.org/pdf/1602.02176.pdf
> * [3]: Fey, Matthias and J. E. Lenssen, Fast Graph Representation Learning with PyTorch Geometric. ArXiv abs/1903.02428 (2019), https://arxiv.org/abs/1903.02428
> * [4]: Morris, C., Ritzert, M., Fey, M., Hamilton, W. L., Lenssen, J. E., Rattan, G., \& Grohe, M. (2019). Weisfeiler and Leman Go Neural: Higher-Order Graph Neural Networks. Proceedings of the AAAI Conference on Artificial Intelligence, 33(01), 4602-4609. https://doi.org/10.1609/aaai.v33i01.33014602
> * [5]: Schlichtkrull, Michael, et al. "Modeling relational data with graph convolutional networks." European semantic web conference. Springer, Cham, 2018. https://arxiv.org/pdf/1703.06103.pdf
> * [6]: Harada, Shonosuke and H. Kashima., GraphITE: Estimating Individual Effects of Graph-structured Treatments., ArXiv abs/2009.14061 (2020): https://arxiv.org/abs/2009.14061

---

### Author Response · Authors · 2021-08-09
**[1/2] Quasi-Oracle Convergence Guarantee**

# Introduction

Since some questions were raised regarding which kind of theoretical guarantees we could provide, in the interest of addressing such questions we post here a new theoretical result we have resolved since the submission of the paper. Specifically, we have derived a new theoretical result on our proposed generalized Robinson decomposition and will add it to the next version. We see this result as first steps towards resolving the lack of theoretical guarantees as mentioned in "_Limitations and future work_" of Section 6. We first state our high-level theoretical result. For the interested reader, we also provide a proof sketch in the second half of the comment.

## Our proposed generalized Robinson decomposition has a quasi-oracle regret bound when using penalized basis-function regression algorithms.

The rates show that as long as the nuisance component $\widehat{m}(\cdot)$ and $\widehat{e}^{h}(\cdot)$ are estimated at $n^{1/4}$ rate, our CATE estimates $\widehat \tau\left(\mathbf{t}^{\prime}, \mathbf{t}, \mathbf{x}\right)$ achieve the same error bounds as an oracle who has a priori knowledge of the true nuisance functionals $m(\cdot)$ and $e^h(\cdot)$. Penalized basis-function regression assumes finite, fixed feature maps $\widehat{g_{\psi}}(\mathbf{X})$, $\widehat{h_{\phi}}(\mathbf{T})$ are enough to represent the true response function; we further assume cross-fitting for the nuisance parameters. Note that both assumptions differ from our more pragmatic training algorithm in Section 4.3 where we learn feature maps simultaenously and have not incorporated cross-fitting. Nonetheless, we believe the result to be immensely useful for practitioners working with the appropriate training procedure (e.g. yielding fixed feature maps through pre-training and adding cross-fitting).

For deriving the result, we extend the quasi-oracle guarantees for the binary treatment setup of [1] to the featurized treatments setting. Our aim is to lay down the first step towards a theoretical foundation for CATE estimation with featurized treatments. We acknowledge that their exist other bounds results for the binary treatment case too, e.g., [2]. Extending them to the featurized treatment setup is an interesting direction for future work.

# Distinctions between binary and featurized treatments
Below we note a few important distinctions between the binary treatment and featurized treatment case.
* In the binary case, we can break the response $Y$ into the CATE $\tau_b(\cdot)$ and the two nuisance functions $m(\cdot)$ and $e^p(\cdot)$. This allows us to treat $\tau_b(\cdot)$ as a weighted regression problem with estimated weights (including the nuisance parameters) and reuse any non-parametric least squares regression rates for the oracle risk bound; as shown in [1]. Meanwhile, in the featurized treatment case we consider, since there are more than two treatments, we must directly analyze the product effect response $f(\mathbf{X}, \mathbf{T})$. The nuisance parameters enter as vectors; we can not factor out the nuisance parameters and view them as weights.

* The overlap condition has a different implication for each case. A key step in achieving the quasi-oracle bound is the ability to bound the difference in function norm of any estimator from the optimal estimator by the regret of that estimator. Since our nuisance estimators enter the regret as a vector, this also makes bounding the function norm more complicated than in the binary case where the nuisance parameters enter as scalars.

# Proof sketch
Now, for the interested readers, here we provide a sketch for our proof. For clarity, we first write down a few definitions. Under the basis representation, a response function $f_{\Theta}$ has representation
$$
(1) \quad\quad	f_{\Theta}\left(\mathbf{X}, \mathbf{T}\right) = \Psi\left(\mathbf{X}\right)^{\top}\Theta \Phi\left(\mathbf{T}\right) \hspace{100cm}
$$

The expected loss function of $\Theta$ is

$$
(2)\quad \quad    	L\left(\Theta\right) = \mathbb{E}\left[\left(\{Y - m^*\left(\mathbf{X}\right)\} - \{\Psi\left(\mathbf{X}\right)^{\top}\Theta \left(\Phi\left(\mathbf{T}\right) - e^p\left(\mathbf{X}\right)\right)\}\right)^2\right], \hspace{10cm}
$$

and the regret of $\Theta$ is defined as the excess risk

$$
(3) \quad\quad	R\left(\Theta\right) = L\left(\Theta\right) - L\left(\Theta^*\right),
\hspace{10cm}
$$

where $\Theta^*$ represents the true response function.

We define the ball of radius $c$, $\mathcal{H}_c := \{\Theta \in \mathbb{R}^d: \|\Theta\|_2 \leq c\}$ where $d$ is the number of basis functions, to be a restricted hypothesis space for $\Theta$, and define

$$
(4)\quad \quad
	\Theta_c := \arg \min_{\Theta \in \mathcal{H}_c} L\left(\Theta\right)
\hspace{10cm}
$$

to be the minimizer of expected loss in $\mathcal{H}_c$. Now we define the expected *c-regret* of $\Theta$ as

$$
(5)\quad \quad
	R\left(\Theta; c\right) = L\left(\Theta\right) - L\left(\Theta_c\right),
\hspace{10cm}
$$

the oracle loss

$$
(6)\quad \quad
	\tilde{L}_n\left(\Theta\right) = \frac{1}{n} \sum_l^n (\{Y_l - m^*(\mathbf{X}_l)\} - \{\Psi(\mathbf{X}_l)^{\top}\Theta (\Phi(\mathbf{T}_l) - e^p(\mathbf{X}_l))\})^2,
\hspace{100cm}
$$

and the feasible loss as

$$
(7)\quad \quad
	\hat{L}_n(\Theta) = \frac{1}{n} \sum_l^n (\{Y - \hat{m}(\mathbf{X})\} - \{\Psi(\mathbf{X})^{\top}\Theta(\Phi(\mathbf{T}) - \hat{e}^p(\mathbf{X}))\})^2,
\hspace{10cm}
$$

where $\hat{m}$ and $\hat{e}^p$ are obtained via cross-fitting.

Analogously, we define $\tilde{R}(\Theta)$, $\hat{R}(\Theta)$, $\tilde{R}(\Theta; c)$, and $\hat{R}(\Theta; c)$.

The goal is to show that the feasible learner (where the nuisance parameters are estimated) satisfies the same expected risk bound as the oracle learner (where the nuisance parameters are provided as oracle quantities).

We follow the strategy of [1], where we first show an almost-isomorphism between the oracle-empirical regret function $\tilde{R}$ and the expected regret function $R$, where by `almost' we mean that the isomorphism can be off by the size of the regularizer. Formally, we need

$$
(8)\quad \quad
	\frac{1}{k}\tilde{R}(f_{\Theta};c) - \rho(c) \leq R(f_{\Theta}; c) \leq k\tilde{R}(f_{\Theta}; c) + \rho(c)
\hspace{10cm}
$$

where $\tilde{R}$ is the oracle-empirical regret. Then a result adapted from [3] immediately implies that given this almost-isomorphism, for any minimizer of the regularised empirical oracle loss


 $$
(9)\quad \quad
	f_{\tilde{\Theta}} \in \operatorname{argmin}_{f \in \mathcal{H}_C}   \tilde{L} (f)+\kappa_1\Lambda(f)
\hspace{10cm}
$$

its expected risk can be bounded by

$$
(10)\quad \quad
	L(f_{\tilde{\Theta}}) \leq \inf_{f \in \mathcal{H}_C} \{L(f) + \kappa_2\Lambda(f)}.
\hspace{10cm}
$$

Then we show that the feasible regret ($\hat{R}$) differs from the oracle regret ($\tilde{R}$) only by a small constant multiple of the expected regret ($R$), i.e.

$$
(11)\quad \quad
\left|\tilde{R}(\Theta; c) - \hat{R}(\Theta; c)\right| \leq 0.25 R(\Theta; c) + o(\rho(c)),
\hspace{10cm}
$$

and therefore $\hat{R}$ also satisfies an almost-isomorphism like (8), just with a different constant $k$. Thus the same expected risk bound holds.

In order to bound the left hand side of (11), we need
1. convergence rates of the nuisance parameters
2. to upper bound the difference in function norm of any estimator $f_{\Theta}$ from the optimal estimator within a $c-$ball $f_{\Theta_{c}}$ by the expected regret of that estimator $R(f_{\Theta};c)$.

For 1., we assume $n^{1/4}$ convergence rates for the nuisance parameters.
For 2., we need to show that:

$$
(12)\quad \quad
	R(\Theta ; c) \geq \epsilon \|f_{\Theta} - f_{\Theta_{c}}\|_{L_2}.
\hspace{10cm}
$$

In order to show (12), we first show by decomposing the square that

$$
   (13) \quad\quad R\left(\Theta; c\right)\hspace{100cm}
$$

$$
	\quad\quad\quad=\mathbb{E}\left[\{f_{\Theta}\left(\mathbf{X}, \mathbf{T}\right) - f_{\Theta_{c}}\left(\mathbf{X}, \mathbf{T}\right) - \mathbb{E}\left[f_{\Theta}\left(\mathbf{X}, \mathbf{T}\right) - f_{\Theta_{c}}\left(\mathbf{X}, \mathbf{T}\right)|\mathbf{X}\right]\}^2\right]
$$

$$
         	\quad\quad\quad\quad+ 2\mathbb{E}\Big[\{\left(f_{\Theta}\left(\mathbf{X}, \mathbf{T}\right) - f_{\Theta_{c}}\left(\mathbf{X}, \mathbf{T}\right)\right) - \mathbb{E}\left[f_{\Theta}\left(\mathbf{X}, \mathbf{T}\right) - f_{\Theta_{c}}\left(\mathbf{X}, \mathbf{T}\right)|\mathbf{X}\right]\}
$$

$$
        	\quad\quad\quad\quad\quad \quad \cdot \big((f_{\Theta_{c}}(\mathbf{X}, \mathbf{T}) - f^*(\mathbf{X}, \mathbf{T})) - \mathbb{E}[f_{\Theta_{c}}(\mathbf{X}, \mathbf{T}) - f^*(\mathbf{X}, \mathbf{T})|\mathbf{X}]\big)\Big].
$$

The second term is always $\geq 0$ due to the convexity of the restricted search space of $\Theta$. The first term can be shown to be equal to

$$
(14)\quad \quad
	\mathbb{E}\Big[\Big\langle \Theta - \Theta_c, \Psi(\mathbf{X}) \otimes \big(\Phi(\mathbf{T}) - e^p(\mathbf{X})\big)\Big\rangle^2\Big]
\hspace{10cm}
$$

by the basis representation of $f_{\Theta}$ and $f_{\Theta_{c}}$. Moreover, (14) can be shown to be lower bounded by the right hand side of (12). To show this we consider the unit sphere of $\Theta$ with norm 1. As long as the feature maps $\Psi(\mathcal{X})$ and $\Phi(\mathcal{T})$ have overlap, strict Jensen's inequality implies that for all $\Theta \in S$, Eq. (14) $>0$. Furthermore, since the search space of $\Theta$ and $\Theta_{c}$ is compact, we know there must be a positive constant $A > 0$ such that (14) $ \geq A$. Then for all $\Theta, \Theta_{c}$ in their search space, we have Eq. (14) $
\geq A||\Theta - \Theta_c||_2$.

Finally, using a lemma connecting $||\Theta - \Theta||_2$  with the $L_2$ difference of their induced functions, we have (12).

One important point to emphasize is that we only require overlap on the feature maps of $\mathbf{X}$ and $\mathbf{T}$, namely $\Psi(\mathbf{X})$ and $\Phi(\mathbf{T})$, and not on the features of the covariates nor the treatments themselves.

---

> ### Author Response · Authors · 2021-08-09
> **[2/2] Quasi-Oracle Convergence Guarantee**
>
> For the rest of the proof, we make use of the structure and machinery of that of [1], which in turns makes heavy use of cross-fitting style arguments. Analogous to them, we replace their $n^{1/4}$ convergence rate assumption of the propensity score (in $\mathbb{R}$) by a $n^{1/4}$ convergence rate of the propensity features in $\mathbb{R}^d$, where $d$ is the number of basis functions.
>
> # References
>
> [1]: X Nie, S Wager, Quasi-oracle estimation of heterogeneous treatment effects, Biometrika, Volume 108, Issue 2, June 2021, Pages 299–319, https://doi.org/10.1093/biomet/asaa076
>
> [2]: Kennedy, Edward H., Optimal doubly robust estimation of heterogeneous causal effects, arXiv e-print 2004.14497, 2020, https://arxiv.org/abs/2004.14497
>
> [3]: P. L. Bartlett. Fast rates for estimation error and oracle inequalities for model selection, Econometric Theory, 24(2):545–552, 2008. https://www.jstor.org/stable/20142503

---

### Decision · Program_Chairs · 2021-09-27

**Decision:**

Accept (Poster)

**Comment:**

The proposed method extends the R-learner (Nie and Wager, 2020) to deal with multiple treatments that are graph-structured. Dealing with non-Euclidean treatments is a very important problem that is understudied in the literature. The proposed two-stage algorithm is an extension of the Robinson decomposition that several reviewers found promising (kq9J, rvHB). An important stepping stone of the approach is to generalize the propensity scores to propensity features. Reviewers rvHB and 1AJd criticized the overlap assumption, which is very strong if the treatment is graph-structured. In the discussion, it has been clarified that the overlap assumption is only required for the features maps of X and T. In addition, the authors provided a sketch for a quasi-oracle regret bound. Reviewers kq9J, rvHB and fdib agree that the paper is well-written. Reviewer fdib pointed to some issues in the presentation. Overall, the authors have addressed the main issues in their responses and the theoretical results have been strengthened during the review process.